# Q-DiT4SR: Exploration of Detail-Preserving Diffusion Transformer Quantization for Real-World Image Super-Resolution

Xun Zhang [1] [*]   Kaicheng Yang [1] [*]   Hongliang Lu [1]   Haotong Qin [2]   Yong Guo [3]   Yulun Zhang [1] [†]

## Abstract

Recently, Diffusion Transformers (DiTs) have emerged in Real-World Image Super-Resolution (Real-ISR) to generate high-quality textures, yet their heavy inference burden hinders real-world deployment. While Post-Training Quantization (PTQ) is a promising solution for acceleration, existing methods in super-resolution mostly focus on U-Net architectures, whereas generic DiT quantization is typically designed for text-to-image tasks. Directly applying these methods to DiT-based super-resolution models leads to severe degradation of local textures. Therefore, we propose **Q-DiT4SR**, the first PTQ framework specifically tailored for DiT-based Real-ISR. We propose **H-SVD**, a hierarchical SVD that integrates a global low-rank branch with a local block-wise rank-1 branch under a matched parameter budget. We further propose **V**ariance-**a**ware **S**patio-**T**emporal **M**ixed **P**recision: **VaSMP** allocates cross-layer weight bit-widths in a data-free manner based on rate-distortion theory, while **VaTMP** schedules intra-layer activation precision across diffusion timesteps via dynamic programming (DP) with minimal calibration. Experiments on multiple real-world datasets demonstrate that our Q-DiT4SR achieves SOTA performance under both **W4A6** and **W4A4** settings. Notably, the W4A4 quantization configuration reduces model size by **5.8×** and computational operations by **6.14×**. Our code and models will be available at *https://github.com/xunzhang1128/Q-DiT4SR*.

## 1. Introduction

Real-world image super-resolution (Real-ISR) aims to restore high-quality images from low-resolution (LR) inputs

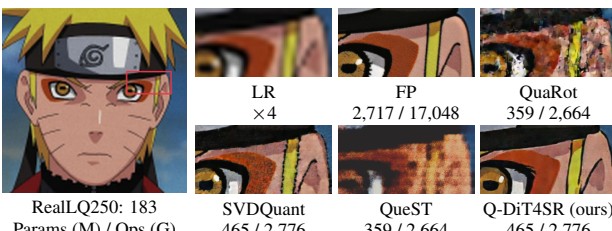

*Figure 1.* Visual comparison (×4) under W4A4 setting. Compared with existing quantization methods, our quantized 4-bit Q-DiT4SR better preserves fine-grained textures, while remaining visually close to the full-precision (FP) model. We also provide the parameter (*i.e.*, Params (M)) and operation numbers (*i.e.*, Ops (G)).

degraded by complex processes, with applications in medical imaging, satellite reconnaissance, and consumer photography (Wang et al., 2021; 2018; Zhang et al., 2021). Traditional methods have evolved from CNNs (Dong et al., 2014; Kim et al., 2016b; Lim et al., 2017; Zhang et al., 2018c) and Transformers (Liang et al., 2021; Liu et al., 2021; Zamir et al., 2022) to diffusion-based models (Rombach et al., 2022; Ho et al., 2020; Song et al., 2021). State-of-the-art diffusion methods, such as StableSR (Wang et al., 2024), DiffBIR (Lin et al., 2024), and SeeSR (Wu et al., 2024c), demonstrate strong restoration capabilities by jointly leveraging semantic guidance and pixel-aware conditions.

Recent Diffusion Transformers (DiTs) (Peebles & Xie, 2023), built entirely on linear layers and self-attention, achieve superior restoration quality (Duan et al., 2025; Ai et al., 2024; Cheng et al., 2025). However, a typical DiT model is characterized by an extremely large parameter count and heavy computational complexity, accompanied by substantial memory overhead. This computational bottleneck limits the practical deployment of DiT-based SR.

Post-training quantization (PTQ) (Li et al., 2021; Hubara et al., 2021; Nagel et al., 2019) compresses weights and activations with light fine-tuning. However, quantizing diffusion models is challenging: quantization errors accumulate across timesteps (Li et al., 2023a; So et al., 2023), activation distributions vary dramatically (He et al., 2023; Shang et al., 2023), and Real-ISR is sensitive to high-frequency distortions. While recent DiT quantization methods (Wu et al., 2024a; Chen et al., 2025; Li et al., 2024; Yang et al., 2025) have made progress, they either require extensive calibration or suffer quality degradation under aggressive bit-widths

---

[*]Equal contribution [1]Shanghai Jiao Tong University [2]ETH Zürich [3]Huawei. Correspondence to: Yulun Zhang[†] <yulun100@gmail.com>.

*Proceedings of the 43rd International Conference on Machine Learning*, Seoul, South Korea. PMLR 306, 2026. Copyright 2026 by the author(s).

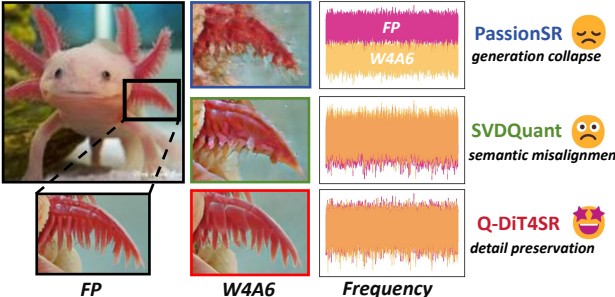

*Figure 2.* **Left**: Visual comparisons (×4) on RealLR200 under W4A6. **Right**: Frequency energy of the *blocks.23.ff.net.2* layer outputs from three W4A6 models, compared with the FP model.

(*e.g.*, W4A4), inadequately addressing three key challenges: **(1)** Many matrix decomposition approaches overly simplify model representations, making it difficult to retain the fine-grained image details required for texture reconstruction. **(2)** Despite pronounced inter-layer differences in DiT-based SR models, current mixed-precision quantization methods lack a calibration-free mechanism to assign different weight bit-widths across layers. **(3)** In diffusion-based Real-ISR, most activation quantization schemes adopt static precision across timesteps, neglecting to exploit temporal dynamics.

To address those challenges, we propose **Q-DiT4SR**, a post-training quantization (PTQ) framework that improves both weight approximation and precision scheduling for efficient DiT-based Real-ISR. We propose **H-SVD** (Hierarchical SVD) that decomposes weights into a global low-rank branch and a local block-wise rank-1 branch under matched parameter budget. We further propose **V**ariance-**a**ware **S**patio-**T**emporal **M**ixed **P**recision: **VaSMP** allocates different weight bits across layers in a data-free manner, while **VaTMP** schedules intra-layer activation precision across timesteps. Experiments on real-world benchmarks demonstrate Q-DiT4SR achieves SOTA performance under W4A6 and W4A4 settings (see Fig. 1 and 2). Notably, our W4A4 quantized model reduces size by **5.8×** and operations by **6.14×** compared with the full-precision (FP) version.

Our main contributions are summarized as follows:

- We propose Q-DiT4SR, a PTQ framework for DiT-based Real-ISR. To the best of our knowledge, our Q-DiT4SR is the first attempt to systematically explore aggressive low-bit PTQ for DiT-based Real-ISR.
- We propose H-SVD, a hierarchical SVD weight decomposition strategy, which significantly improves the preservation of fine-grained textures critical for Real-ISR under aggressive low-bit quantization.
- We propose variance-aware spatio-temporal mixed precision: VaSMP for calibration-free cross-layer weight precision allocation, and VaTMP for intra-layer activation precision scheduling across diffusion timesteps.
- Extensive experiments on multiple real-world datasets demonstrate that our Q-DiT4SR achieves SOTA performance under both W4A6 and W4A4 settings.

## 2. Related Work

### 2.1. Image Super-Resolution

Deep learning-based image super-resolution (SR) has evolved from early convolutional neural networks (CNNs) (Dong et al., 2014; Kim et al., 2016b;a) to architectures with residual learning (He et al., 2016; Lim et al., 2017; Zhang et al., 2018a;c) and Transformers (Liang et al., 2021; Chen et al., 2023a;b). For Real-World Image Super-Resolution (Real-ISR) , GAN-based methods (Wang et al., 2018; 2021; Liang et al., 2021) pioneer perceptually realistic restoration, while recent breakthroughs leverage latent diffusion models (Rombach et al., 2022). State-of-the-art diffusion-based SR methods, such as StableSR (Wang et al., 2024), DiffBIR (Lin et al., 2024), SeeSR (Wu et al., 2024c) and OSEDiff (Wu et al., 2024b) achieve strong quality via semantic guidance and pixel-aware conditioning. Recent work has demonstrated that Diffusion Transformer (DiT) architectures, such as DiT4SR (Duan et al., 2025) and Dream-Clear (Ai et al., 2024), achieve impressive performance. Notably, traditional SR models rely heavily on convolutional layers, and most existing quantization methods are tailored for such architectures. In contrast, recent DiT-based SR models depend mostly on linear layers, presenting new challenges for efficient low-bit quantization.

### 2.2. Quantization of Diffusion Models

Quantizing diffusion models is challenging due to iterative denoising and activation outliers across timesteps. Quantization-aware training (QAT) approaches (Zhou et al., 2016; Esser et al., 2019; Qin et al., 2023; Zheng et al., 2024; 2025; Yang et al., 2025; Li et al., 2023b) can fine-tune models for low-bit precision but require expensive retraining and large-scale datasets. In contrast, post-training quantization (PTQ) enables deployment with light fine-tuning. Early PTQ methods for general diffusion models, such as Q-Diffusion (Li et al., 2023a) and PTQD (He et al., 2023), adapt calibration strategies for timestep-dependent distributions, while Temporal Dynamic Quantization (So et al., 2023) introduces timestep-aware bit allocation. For Diffusion Transformers (DiTs), PTQ4DiT (Wu et al., 2024a) proposes channel reordering and block reconstruction, and Q-DiT (Chen et al., 2025) introduces window attention-aware rounding. SVDQuant (Li et al., 2024) employs low-rank decomposition to mitigate outliers, enabling 4-bit quantization. RobuQ (Yang et al., 2025) leverages Hadamard transforms for W1.58A2 quantization in DiTs. For SR-specific diffusion models, PassionSR (Zhu et al., 2025) tunes adaptive scales in one-step models. However, most PTQ methods require much calibration data and iterative optimization. In contrast, our approach achieves training-free PTQ with minimal calibration overhead: VaSMP allocates weight precision offline without any calibration, while VaTMP requires only a tiny calibration set for activation precision allocation.

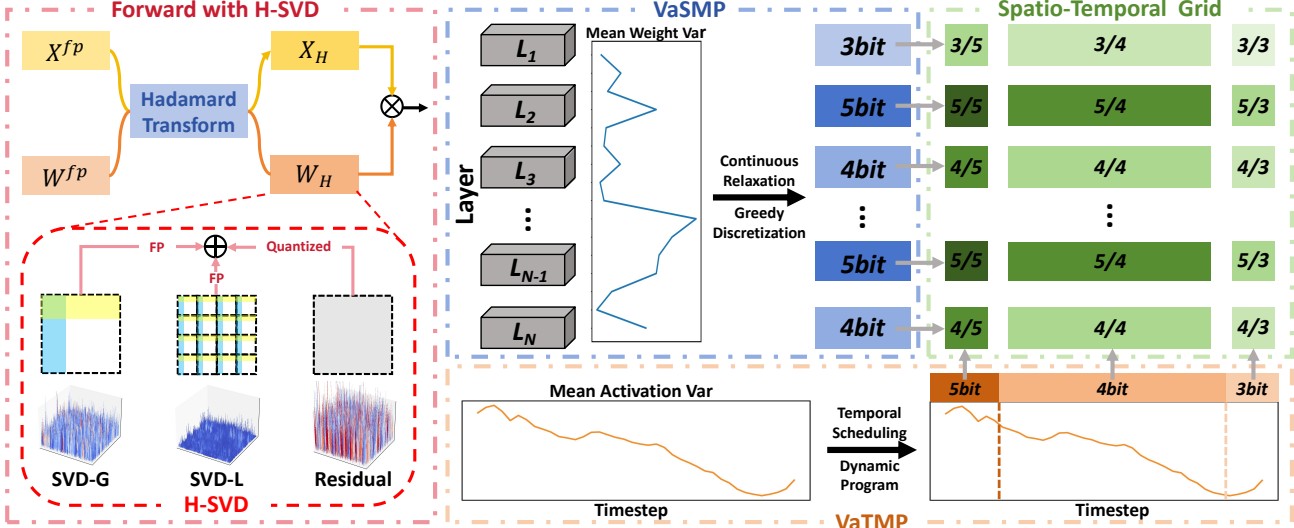

*Figure 3.* Overview of our proposed Q-DiT4SR framework. (a) Forward with H-SVD: we reconstruct and quantize DiT weights using a hierarchical decomposition that integrates SVD-G with SVD-L. (b) VaSMP: a cross-layer mixed-precision assignment for weights driven by mean weight variance. (c) VaTMP: a intra-layer scheduling that allocates activation precision across diffusion timesteps using mean activation variance. The spatio-temporal grid specifies per-(layer, timestep) bit configurations (weight/activation bit pairs) for deployment.

### 2.3. Mixed-Precision Quantization

Mixed-precision quantization allocates different bit-widths per layer or component to balance accuracy and efficiency. HAWQ (Dong et al., 2019) and its successors (Dong et al., 2020; Yao et al., 2021) use Hessian-based sensitivity for bit allocation, while BRECQ (Li et al., 2021) performs block-wise reconstruction. Recent work extends to Transformers and diffusion models: MixDQ (Zhao et al., 2024) decouples metric-based sensitivity for few-step text-to-image models, HQ-DiT (Liu & Zhang, 2024) introduces FP4 hybrid quantization, and MPQ-DM (Feng et al., 2025a) as well as MPQ-DMv2 (Feng et al., 2025b) applies residual mixed-precision with temporal distillation. For large language models (LLMs), CoopQ (Zhao et al., 2025) and AMQ (Lee et al., 2025) enable automated mixed-precision via interaction-aware or AutoML-based strategies. However, most methods require expensive calibration with large datasets or iterative forward passes. In contrast, our VaSMP allocates weight precision offline without any calibration, while VaTMP uses only a tiny calibration set with negligible computation to determine activation precision via variance statistics.

## 3. Method

### 3.1. Preliminaries

**Activation Quantization.** Given activations $\mathbf{X} \in \mathbb{R}^{T \times C}$, each token $\mathbf{x} = \mathbf{X}_{t,:}^{\top} \in \mathbb{R}^{C}$ is transformed using a normalized Hadamard matrix $\mathbf{H}_n$ (Ashkboos et al., 2024; Yang et al., 2025; Tseng et al., 2024; Liu et al., 2025b): $\mathbf{Z} = \mathbf{X}\mathbf{H}_n$. Under some assumptions, the transformed activations are approximately Gaussian on a per-token basis,

$$\mathbf{Z}_{t,:} \approx \mathcal{N}\left(\mathbf{0},\, \sigma_t^2 \mathbf{I}\right), \qquad \sigma_t^2 = \frac{1}{C} \left\| \mathbf{Z}_{t,:} \right\|_2^2. \quad (1)$$

where $\sigma_t$ is estimated per token (Yang et al., 2025; Kolb et al., 2023). Quantization is performed using a symmetric uniform quantizer $Q_{\mathrm{uni}}(\cdot)$ pre-optimized for $\mathcal{N}(0, 1)$:

$$Q_G(\mathbf{x}) = \sigma_t \cdot Q_{\mathrm{uni}}\left(\frac{\mathbf{H}_n^{\top} \mathbf{x}}{\sigma_t}\right). \quad (2)$$

**Weight Quantization.** For a weight matrix $\mathbf{W} \in \mathbb{R}^{out \times in}$, a normalized Hadamard transform is also applied (Ashkboos et al., 2024; Yang et al., 2025; Tseng et al., 2024; Liu et al., 2025b): $\mathbf{W}_{\mathrm{H}} = \mathbf{W}\mathbf{H}_n$. A truncated SVD is then computed to retain a rank-$r$ FP low-rank branch $\mathbf{W}_{\mathrm{LRB}}$ (*e.g.*, $r{=}32$) (Yang et al., 2025; Li et al., 2024; Liu et al., 2025a). The residual $\mathbf{W}_{\mathrm{res}} = \mathbf{W}_{\mathrm{H}} - \mathbf{W}_{\mathrm{LRB}}$ is then quantized using the same uniform quantizer as used for activations:

$$Q_w(\mathbf{W}_{\mathrm{res}}) = \sigma_o \cdot Q_{\mathrm{uni}}\left(\frac{\mathbf{W}_{\mathrm{res}}}{\sigma_o}\right), \quad (3)$$

where $\sigma_o$ is estimated independently for each output channel. The final weight is then quantized and reconstructed as (Li et al., 2024; Yang et al., 2025; Liu et al., 2025a),

$$\hat{\mathbf{W}} = \left(\mathbf{W}_{\mathrm{LRB}} + Q_w(\mathbf{W}_{\mathrm{res}})\right) \mathbf{H}_n^{\top}. \quad (4)$$

Overall, this framework makes both weights and activations approximately Gaussian, enabling efficient and stable PTQ.

### 3.2. Hierarchical SVD

#### 3.2.1. MOTIVATION

Diffusion-based image SR models are highly sensitive to high-frequency details. Under the quantization backbone (Sec. 3.1), each weight matrix is decomposed into a FP low-rank branch and a quantized residual. While the low-rank component captures dominant global structures, the residual often contains critical high-frequency and local information, making it particularly vulnerable to low-bit quantization.

To better preserve FP information flow, we propose a hierarchical SVD (H-SVD) that integrates a complementary global low-rank branch with a local block-wise rank-1

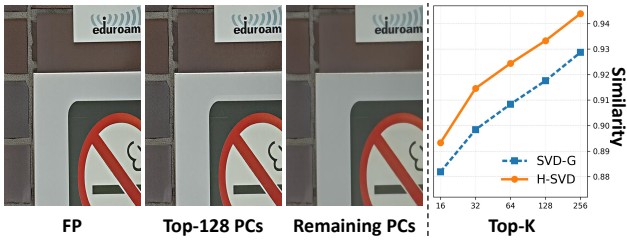

*Figure 4.* PCA analysis on RealSR Canon_001. **Left**: Top-128 principal components (PCs) are progressively removed from the output of the *blocks.23.ff.net.2* layer in the FP model. Removing dominant PCs leads to a significant degradation in SR quality, highlighting their importance in SR tasks. **Right**: Comparison between SVD-G and H-SVD under matched parameter budgets for the same non-quantized layer, with outputs projected onto the FP PC space. The hierarchical decomposition aligns more closely with the FP model, demonstrating the effectiveness of H-SVD.

branch in the transformed weight space (Fig. 3 (left)). By jointly modeling global and local structures, H-SVD enables a more faithful approximation of FP weights under a fixed parameter budget. Empirically, as shown in Fig. 4 (left), a small number of principal components (PCs) dominate image SR quality, motivating the allocation of adaptation capacity across both global and local SVD components.

### 3.2.2. GLOBAL SVD CONSTRUCTION

Following Sec. 3.1, we first apply a Hadamard transform to the weight matrix to obtain a more balanced distribution: $\mathbf{W}_{\mathrm{H}} = \mathbf{W}\mathbf{H}_n$. We then perform a truncated SVD on $\mathbf{W}_{\mathrm{H}}$ to extract a FP global approximation, denoted as $\mathbf{W}_{\mathrm{SVD\text{-}G}}$. This global branch (SVD-G) captures the dominant low-frequency components of the weight matrix, which are crucial for preserving the overall structural behavior of the FP model. The remaining residual is defined as $\mathbf{W}_{\mathrm{res}} = \mathbf{W}_{\mathrm{H}} - \mathbf{W}_{\mathrm{SVD\text{-}G}}$, which primarily contains fine-grained and local information that is difficult to approximate using a single global low-rank branch alone.

### 3.2.3. LOCAL SVD CONSTRUCTION

Given the residual weight $\mathbf{W}_{\mathrm{res}} \in \mathbb{R}^{out \times in}$, we construct a local SVD branch (SVD-L) by partitioning $\mathbf{W}_{\mathrm{res}}$ into some non-overlapping blocks of size $s_o \times s_i$. This yields $B = \frac{out}{s_o} \cdot \frac{in}{s_i}$ small blocks $\mathbf{W}^{(p,q)} \in \mathbb{R}^{s_o \times s_i}$.

For each block, we apply a rank-1 SVD approximation
$$\mathbf{W}^{(p,q)} \approx \hat{\mathbf{W}}^{(p,q)} = \sigma_{p,q}\, \mathbf{u}_{p,q}\mathbf{v}_{p,q}^{\top}, \qquad (5)$$
where $\mathbf{u}_{p,q} \in \mathbb{R}^{s_o}$, $\mathbf{v}_{p,q} \in \mathbb{R}^{s_i}$, and $\sigma_{p,q} \in \mathbb{R}_+$ denote the top singular components. All block-wise approximations are assembled to form the local SVD branch
$$\mathbf{W}_{\mathrm{SVD\text{-}L}} = \mathrm{Assemble}\Big(\{\hat{\mathbf{W}}^{(p,q)}\}_{p,q}\Big) \in \mathbb{R}^{out \times in}. \quad (6)$$
Let $r$ denote the target rank used as a unified budget reference. A standard global SVD branch parameterizes $\mathbf{W}_{\mathrm{SVD\text{-}G}} = \mathbf{A}\mathbf{B}$ with $\mathbf{A} \in \mathbb{R}^{out \times r}$ and $\mathbf{B} \in \mathbb{R}^{r \times in}$, leading to a parameter budget $P_{\mathrm{SVD\text{-}G}}(r) = r\,(out + in)$. A rank-1 block $\hat{\mathbf{W}}^{(p,q)}$ requires $P_{\mathrm{blk}}(s_o, s_i) = s_o + s_i + 1$

parameters, and the total budget of the local branch is
$$P_{\mathrm{SVD\text{-}L}}(s_o, s_i) = \frac{out}{s_o} \cdot \frac{in}{s_i} \cdot (s_o + s_i + 1). \qquad (7)$$
We search over all feasible block sizes
$$\mathcal{S} = \{(s_o, s_i) :\ s_o \mid out,\ s_i \mid in\}, \qquad (8)$$
and select a configuration satisfying the constraint
$$P_{\mathrm{SVD\text{-}L}}(s_o, s_i) \ \lesssim\ P_{\mathrm{SVD\text{-}G}}(r), \qquad (9)$$
ensuring that SVD-L preserves fine-grained textures while using a parameter budget comparable to rank-$r$ SVD-G.

### 3.2.4. FORWARD COMPUTATION WITH H-SVD
The final weight is reconstructed with H-SVD:
$$\begin{aligned}\hat{\mathbf{W}} =&\ (\mathbf{W}_{\mathrm{SVD\text{-}G}} + \mathbf{W}_{\mathrm{SVD\text{-}L}})\,\mathbf{H}_n^{\top} \\ &+ Q_w(\mathbf{W}_{\mathrm{res}} - \mathbf{W}_{\mathrm{SVD\text{-}L}})\,\mathbf{H}_n^{\top}.\end{aligned} \qquad (10)$$
Accordingly, the forward computation under quantization is
$$\begin{aligned}\mathbf{y} =&\ Q_w(\mathbf{W}_{\mathrm{res}} - \mathbf{W}_{\mathrm{SVD\text{-}L}})\,Q_G(\mathbf{x}) \\ &+ (\mathbf{W}_{\mathrm{SVD\text{-}G}} + \mathbf{W}_{\mathrm{SVD\text{-}L}})\,\mathbf{H}_n^{\top}\mathbf{x}.\end{aligned} \qquad (11)$$
By allocating the same parameter budget as a rank-$r$ global SVD branch, H-SVD leverages block-wise rank-1 local structures to better preserve critical residual information. As shown in Fig. 4 (right), $\mathbf{W}_{\mathrm{SVD\text{-}L}}$ retains the dominant components and aligns more closely with the FP model.

## 3.3. Variance-Aware Spatio Mixed Precision

### 3.3.1. MOTIVATION
Consider a scalar random variable $z \sim \mathcal{N}(0, \sigma^2)$ quantized by a $b$-bit symmetric uniform quantizer. Under the standard high-rate approximation, the quantization error satisfies
$$\mathbb{E}[e^2] \ \propto\ \sigma^2\, 2^{-2b}, \qquad (12)$$
where the step size scales as $\Delta = \mathcal{O}(\sigma\, 2^{-b})$ for a Gaussian source clipped to $[-\alpha\sigma, \alpha\sigma]$.

Extending to a Hadamard-transformed weight matrix $\mathbf{W}_{\mathrm{H}}^{(\ell)}$ in layer $\ell$, let $\mathbf{w}_{\ell,o}$ denote its $o$-th output-channel row with variance $\sigma_{\ell,o}^2$. Elementwise uniform quantization with bit-width $b_\ell$ induces a layer-wise distortion
$$D_\ell(b_\ell) \triangleq \mathbb{E}\Big[\|\mathbf{Q}(\mathbf{W}_{\mathrm{H}}^{(\ell)}) - \mathbf{W}_{\mathrm{H}}^{(\ell)}\|_F^2\Big] \ \propto\ N_\ell\, \bar{\sigma}_\ell^2\, 2^{-2b_\ell}, \quad (13)$$
where $N_\ell$ is the number of parameters in layer $\ell$ and $\bar{\sigma}_\ell^2 = \frac{1}{out_\ell} \sum_{o=1}^{out_\ell} \sigma_{\ell,o}^2$ is the average output-channel variance.

Eq. (13) indicates that, under a shared uniform-quantization design, the expected layer distortion is dominated by $\bar{\sigma}_\ell^2$. In Fig. 5, weight variances exhibit strong inter-layer diversity and remain relatively stable within a layer. It motivates an optimal data-free variance-aware spatio mixed precision (VaSMP) strategy that allocates weight bit-widths based on $\bar{\sigma}_\ell^2$ without any calibration data, as shown in Fig. 3 (middle).

### 3.3.2. BIT ALLOCATION WITH VASMP

**Offline Statistics.** All statistics are collected offline without any calibration data. For each layer $\ell$, we compute the average output-channel variance $\bar{\sigma}_\ell^2$ and parameter count $N_\ell$

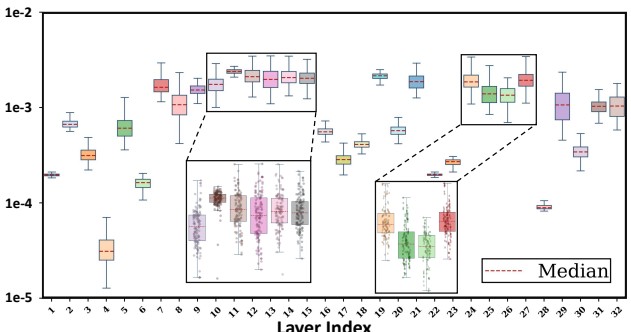

*Figure 5.* Variance Distribution Analysis. The boxplot is generated by randomly selecting 32 different types of layers and calculating the variance of 128 randomly selected output channels for each layer. The variance of Hadamard-transformed weights varies by orders of magnitude across layers, while remaining relatively stable across output channels within a single layer.

in the Hadamard domain. We define the active set $\mathcal{A} \triangleq \{\ell : \text{layer } \ell \text{ participates in the forward computation}\}$.

**Continuous Relaxation.** Motivated by Eq. (13), we minimize the distortion under a target average bit-width $B_{\text{target}}$:

$$\min_{\{b_\ell\}_{\ell \in \mathcal{A}}} \sum_{\ell \in \mathcal{A}} N_\ell \bar{\sigma}_\ell^2 2^{-2b_\ell}$$
$$\text{s.t.} \quad \sum_{\ell \in \mathcal{A}} w_\ell b_\ell = B_{\text{target}} \sum_{\ell \in \mathcal{A}} w_\ell, \quad (14)$$

where $w_\ell = N_\ell$. Let

$$\overline{\log_2 \bar{\sigma}} = \frac{\sum_{\ell \in \mathcal{A}} w_\ell \log_2(\max(\bar{\sigma}_\ell^2, \epsilon))}{\sum_{\ell \in \mathcal{A}} w_\ell}, \quad (15)$$

then the optimal continuous solution is

$$b_\ell^* = B_{\text{target}} + \tfrac{1}{2}\left(\log_2(\max(\bar{\sigma}_\ell^2, \epsilon)) - \overline{\log_2 \bar{\sigma}}\right), \ell \in \mathcal{A}. \quad (16)$$

**Greedy Discretization.** We discretize $\{b_\ell^*\}$ by initializing $b_\ell \leftarrow \text{clip}(\lfloor b_\ell^* \rfloor, b_{\min}, b_{\max})$ and greedily allocating remaining bits. From Eq. (13), increasing $b_\ell$ by one bit reduces the distortion proportionally to $\text{Gain}_\ell \propto \bar{\sigma}_\ell^2 4^{-b_\ell}$, which serves as the priority score for bit assignment. The resulting integer bit-widths are used for inference.

### 3.4. Variance-Aware Temporal Mixed Precision

#### 3.4.1. MOTIVATION

Due to different layer families in diffusion SR backbones, cross-layer activation mixed precision is unreliable. After the Hadamard transform, token activations are approximately Gaussian and exhibit a clear temporal variance trend (Fig. 6 (left)). Since uniform quantization distortion scales with variance (Eq. (12)), timestep-wise variance naturally serves as an intra-layer sensitivity indicator. As shown in Fig. 3 (right lower), we therefore propose a variance-aware temporal mixed precision (VaTMP) strategy for activations.

Formally, for token activations $\mathbf{X}^{(\ell,t)} \in \mathbb{R}^{N \times C}$ at layer $\ell$ and diffusion timestep $t$, we apply $\mathbf{Z}^{(\ell,t)} = \mathbf{X}^{(\ell,t)} \mathbf{H}_n$, and define the timestep sensitivity as the mean token variance

$$v_{\ell,t} \triangleq \frac{1}{N \cdot C} \sum_{n=1}^{N} \|\mathbf{z}_n^{(\ell,t)}\|_2^2, \quad (17)$$

where $\mathbf{z}_n^{(\ell,t)} \in \mathbb{R}^C$ denotes the $n$-th token at timestep $t$.

#### 3.4.2. BIT ALLOCATION WITH VaTMP

We perform timestep-wise activation mixed-precision allocation within each layer after fixing the weight precision (Sec. 3.3, with H-SVD in Sec. 3.2 if enabled). The goal is to minimize activation quantization distortion across diffusion timesteps under a fixed average activation-bit budget.

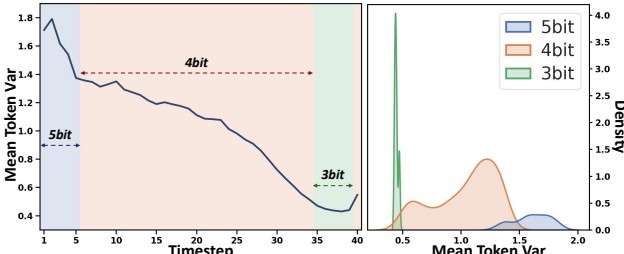

*Figure 6.* Activation variance and bit-width allocations across diffusion timesteps. **Left**: Evolution of the mean token-wise activation variance over diffusion timesteps, measured on the activations of the *block11.attn2.to_v_control* layer. Background colors indicate the timestep-wise bit-width assignment. **Right**: Kernel density estimation (KDE) of the mean token-wise activation variance, where the shaded area reflects the proportion of timesteps.

**Distortion Modeling.** Consider a scalar activation element $z$ (one channel of one token) at layer $\ell$ and diffusion timestep $t$ after the Hadamard transform. We quantize $z$ using a symmetric uniform quantizer with clipping threshold $A > 0$ and bit-width $b \in \mathcal{B}$ (e.g., $\mathcal{B} = \{2, 3, 4, 5, 6, 7, 8\}$):

$$Q_{A,b}(z) = \Delta \cdot \left\lfloor \frac{\text{clip}(z, -A, A)}{\Delta} \right\rceil, \Delta = \frac{2A}{2^b - 1}. \quad (18)$$

Here $\text{clip}(z, -A, A) = \min(\max(z, -A), A)$ and $\lfloor \cdot \rceil$ denotes rounding-to-nearest. Following the Gaussian assumption (Sec. 3.1), we model $z \sim \mathcal{N}(0, v_{\ell,t})$. Let $Z \sim \mathcal{N}(0, 1)$ and write $z = \sqrt{v_{\ell,t}} Z$. We define a normalized distortion coefficient on $\mathcal{N}(0, 1)$ with optimal clipping:

$$\kappa(b) \triangleq \min_{A > 0} \mathbb{E}_{Z \sim \mathcal{N}(0,1)}\left[(Z - Q_{A,b}(Z))^2\right], b \in \mathcal{B}, \quad (19)$$

and denote the corresponding optimal normalized clipping threshold as $A^\star(b) = \arg\min_{A > 0} \mathbb{E}[(Z - Q_{A,b}(Z))^2]$. In practice, for $z \sim \mathcal{N}(0, v_{\ell,t})$, we set the clipping threshold to $A = \sqrt{v_{\ell,t}} A^\star(b)$. Then the expected distortion satisfies

$$D_{\ell,t}(b) \triangleq \mathbb{E}\left[(z - Q_{\sqrt{v_{\ell,t}} A^\star(b), b}(z))^2\right] = v_{\ell,t} \kappa(b). \quad (20)$$

**Temporal Scheduling.** For each layer $\ell$, we assign timestep-wise activation bits $\{b_{\ell,t}\}_{t=1}^{T_\ell}$ under an average-bit budget

$$\sum_{t=1}^{T_\ell} b_{\ell,t} \leq B_\ell \triangleq \left\lfloor B_{\text{target}}^{\text{act}} T_\ell \right\rfloor. \quad (21)$$

We need to minimize the total distortion

$$\min_{\{b_{\ell,t} \in \mathcal{B}\}} \sum_{t=1}^{T_\ell} \kappa(b_{\ell,t}) v_{\ell,t}, \quad \text{s.t.} \sum_{t=1}^{T_\ell} b_{\ell,t} \leq B_\ell, \quad (22)$$

while restricting the solution to piecewise-constant schedules over continuous timestep segments. For a segment $[i, j)$

*Table 1.* Quantitative comparison on four real-world benchmarks. Best and second best performance are highlighted in red and blue.

(a) W4A6 setting

| Datasets | Metrics | FP | Q-Diffusion ICCV 2023 | EfficientDM ICLR 2024 | PTQ4DiT NeurIPS 2024 | QuaRot NeurIPS 2024 | SVDQuant ICLR 2025 | Q-DiT CVPR 2025 | PassionSR CVPR 2025 | FlatQuant ICML 2025 | QueST ICCV 2025 | Q-DiT4SR |
|---|---|---|---|---|---|---|---|---|---|---|---|---|
| DrealSR | LPIPS ↓ | 0.3897 | 0.7045 | 0.7049 | 0.4627 | 0.3925 | 0.3829 | 0.6843 | 0.4774 | 0.4921 | 0.8210 | 0.3880 |
| | MUSIQ ↑ | 64.69 | 57.75 | 59.42 | 56.83 | 61.09 | 61.51 | 57.01 | 50.55 | 53.74 | 40.11 | 64.32 |
| | MANIQA ↑ | 0.4483 | 0.4185 | 0.5027 | 0.3457 | 0.3875 | 0.3931 | 0.4159 | 0.3041 | 0.3296 | 0.3541 | 0.4378 |
| | ClipIQA ↑ | 0.5555 | 0.4200 | 0.3985 | 0.4060 | 0.4876 | 0.4885 | 0.3806 | 0.3705 | 0.3951 | 0.3287 | 0.5492 |
| | LIQE ↑ | 4.031 | 1.928 | 2.057 | 2.579 | 3.275 | 3.344 | 1.709 | 2.036 | 2.340 | 1.145 | 3.930 |
| RealSR | LPIPS ↓ | 0.3179 | 0.7106 | 0.7063 | 0.3928 | 0.3324 | 0.3371 | 0.6855 | 0.4218 | 0.4345 | 0.8656 | 0.3386 |
| | MUSIQ ↑ | 67.89 | 59.63 | 61.26 | 59.77 | 64.47 | 66.63 | 59.02 | 54.85 | 57.11 | 45.99 | 67.72 |
| | MANIQA ↑ | 0.4620 | 0.4122 | 0.5117 | 0.3496 | 0.4041 | 0.4306 | 0.4344 | 0.3096 | 0.3337 | 0.3624 | 0.4566 |
| | ClipIQA ↑ | 0.5482 | 0.4348 | 0.4077 | 0.4239 | 0.4963 | 0.5180 | 0.4002 | 0.3830 | 0.4143 | 0.3350 | 0.5671 |
| | LIQE ↑ | 3.988 | 1.972 | 2.192 | 2.656 | 3.323 | 3.434 | 1.790 | 2.176 | 2.455 | 1.047 | 3.980 |
| RealLR200 | MUSIQ ↑ | 70.33 | 57.87 | 58.80 | 63.25 | 67.51 | 68.78 | 56.78 | 58.18 | 59.86 | 37.46 | 70.36 |
| | MANIQA ↑ | 0.4636 | 0.4067 | 0.5175 | 0.3469 | 0.4121 | 0.4240 | 0.4053 | 0.3001 | 0.3200 | 0.2910 | 0.4635 |
| | ClipIQA ↑ | 0.5814 | 0.4247 | 0.4350 | 0.4544 | 0.5226 | 0.5366 | 0.4243 | 0.4096 | 0.4365 | 0.3232 | 0.5851 |
| | LIQE ↑ | 4.303 | 1.898 | 2.199 | 2.831 | 3.743 | 3.922 | 1.590 | 2.360 | 2.655 | 1.032 | 4.312 |
| RealLQ250 | MUSIQ ↑ | 71.70 | 58.91 | 59.60 | 63.95 | 68.95 | 69.81 | 58.19 | 58.82 | 61.10 | 36.99 | 71.46 |
| | MANIQA ↑ | 0.4614 | 0.4083 | 0.5218 | 0.3284 | 0.4078 | 0.4145 | 0.4047 | 0.2958 | 0.3116 | 0.3104 | 0.4579 |
| | ClipIQA ↑ | 0.5744 | 0.4272 | 0.4385 | 0.4258 | 0.5136 | 0.5175 | 0.4280 | 0.3961 | 0.4229 | 0.3329 | 0.5724 |
| | LIQE ↑ | 4.332 | 1.968 | 2.277 | 2.806 | 3.847 | 3.936 | 1.624 | 2.317 | 2.609 | 1.035 | 4.345 |

(b) W4A4 setting

| Datasets | Metrics | FP | Q-Diffusion ICCV 2023 | EfficientDM ICLR 2024 | PTQ4DiT NeurIPS 2024 | QuaRot NeurIPS 2024 | SVDQuant ICLR 2025 | Q-DiT CVPR 2025 | PassionSR CVPR 2025 | FlatQuant ICML 2025 | QueST ICCV 2025 | Q-DiT4SR |
|---|---|---|---|---|---|---|---|---|---|---|---|---|
| DrealSR | LPIPS ↓ | 0.3897 | 0.7113 | 0.7091 | 0.6970 | 0.5747 | 0.3821 | 0.6748 | 0.6900 | 0.6904 | 0.8322 | 0.4327 |
| | MUSIQ ↑ | 64.69 | 58.44 | 57.44 | 59.26 | 54.47 | 60.65 | 57.75 | 57.48 | 58.39 | 40.05 | 61.86 |
| | MANIQA ↑ | 0.4483 | 0.4991 | 0.4921 | 0.4949 | 0.3357 | 0.3768 | 0.4228 | 0.4684 | 0.4728 | 0.3515 | 0.4030 |
| | ClipIQA ↑ | 0.5555 | 0.3963 | 0.3704 | 0.3968 | 0.3506 | 0.4727 | 0.3953 | 0.3725 | 0.3991 | 0.3253 | 0.4894 |
| | LIQE ↑ | 4.031 | 1.904 | 1.917 | 1.960 | 1.807 | 3.154 | 1.856 | 1.863 | 1.949 | 1.156 | 3.197 |
| RealSR | LPIPS ↓ | 0.3179 | 0.7107 | 0.7126 | 0.6934 | 0.5384 | 0.3268 | 0.6806 | 0.6885 | 0.6871 | 0.8663 | 0.3665 |
| | MUSIQ ↑ | 67.89 | 60.20 | 59.84 | 60.34 | 57.13 | 63.14 | 59.97 | 58.95 | 59.41 | 45.74 | 66.36 |
| | MANIQA ↑ | 0.4620 | 0.5082 | 0.5023 | 0.5104 | 0.3302 | 0.3864 | 0.4310 | 0.4802 | 0.4741 | 0.3612 | 0.4367 |
| | ClipIQA ↑ | 0.5482 | 0.4036 | 0.3748 | 0.4050 | 0.3470 | 0.4729 | 0.4064 | 0.3831 | 0.4083 | 0.3242 | 0.4956 |
| | LIQE ↑ | 3.988 | 2.003 | 2.039 | 2.141 | 1.844 | 3.115 | 2.009 | 1.958 | 1.996 | 1.050 | 3.179 |
| RealLR200 | MUSIQ ↑ | 70.33 | 57.95 | 57.35 | 57.63 | 56.34 | 67.37 | 58.16 | 55.96 | 56.47 | 37.21 | 68.98 |
| | MANIQA ↑ | 0.4636 | 0.5111 | 0.5141 | 0.5070 | 0.3010 | 0.4028 | 0.4267 | 0.4626 | 0.4460 | 0.2897 | 0.4265 |
| | ClipIQA ↑ | 0.5814 | 0.4257 | 0.4118 | 0.4204 | 0.3754 | 0.5168 | 0.4266 | 0.3948 | 0.4162 | 0.3122 | 0.5293 |
| | LIQE ↑ | 4.303 | 2.032 | 2.050 | 2.130 | 1.963 | 3.731 | 1.959 | 1.950 | 1.973 | 1.033 | 3.827 |
| RealLQ250 | MUSIQ ↑ | 71.70 | 58.46 | 58.03 | 58.70 | 57.33 | 69.05 | 59.40 | 57.09 | 57.76 | 36.87 | 69.80 |
| | MANIQA ↑ | 0.4614 | 0.5154 | 0.5190 | 0.5153 | 0.3034 | 0.3939 | 0.4279 | 0.4720 | 0.4566 | 0.3084 | 0.3943 |
| | ClipIQA ↑ | 0.5744 | 0.4297 | 0.4166 | 0.4348 | 0.3628 | 0.5053 | 0.4245 | 0.3970 | 0.4194 | 0.3318 | 0.5026 |
| | LIQE ↑ | 4.332 | 2.111 | 2.118 | 2.186 | 1.985 | 3.803 | 2.029 | 2.018 | 2.080 | 1.042 | 3.766 |

assigned bit-width $b$, the segment cost is

$$\text{SegCost}_\ell(i, j; b) = \kappa(b) \sum_{t=i}^{j-1} v_{\ell, t}. \quad (23)$$

We collect $\{v_{\ell, t}\}$ on a small low-resolution (LR) calibration set under quantized weights. We solve the resulting segmented scheduling problem via dynamic programming (DP) and deploy the inferred per-timestep activation-bit schedule. This procedure assigns higher precision to high-variance (more sensitive) timesteps and lower precision to low-variance ones, while satisfying the same average activation-bit budget (*e.g.*, 4bits), as shown in Fig. 6 (right).

## 4. Experiments

### 4.1. Settings

We adopt DiT4SR (Duan et al., 2025) as the backbone for all experiments. Quantization and evaluation are performed on top of this model, where all MM-DiT blocks are quantized, while softmax layers are fixed at 8-bit precision for numerical stability. All experiments are conducted at a ×4

scaling factor and run on a single NVIDIA RTX A6000 GPU for both calibration and evaluation.

**Calibration Dataset.** We use an LR-only calibration set randomly sampled from the RealSR training split (32 images, random $128 \times 128$ crops), and collect activation statistics under the quantized inference pipeline.

**Test Datasets.** Following DiT4SR, we evaluate the models on four widely used real-world benchmarks: DrealSR (Wei et al., 2020), RealSR (Cai et al., 2019), RealLR200 (Wu et al., 2024c), and RealLQ250 (Ai et al., 2024).

**Metrics.** Following DiT4SR, we use both full- and no-reference metrics for evaluation. Specifically, we adopt LPIPS (Zhang et al., 2018b) as a full-reference perceptual metric, and MUSIQ (Ke et al., 2021), MANIQA (Yang et al., 2022), CLIPIQA (Wang et al., 2023), and LIQE (Zhang et al., 2023) as no-reference quality metrics.

**Compared Methods.** For PTQ, we select multiple methods: SVDQuant (Li et al., 2024), QuaRot (Ashkboos et al., 2024)

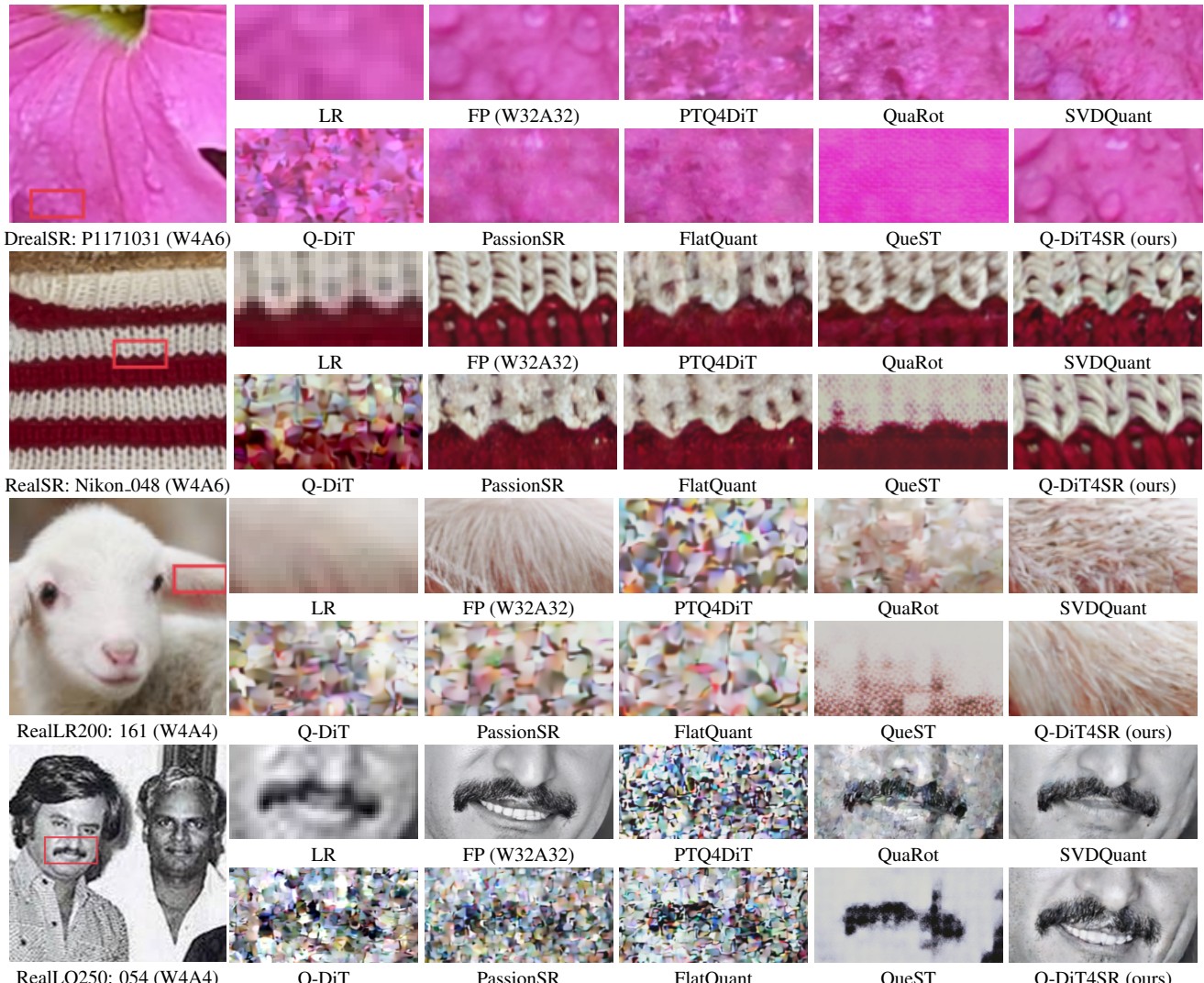

*Figure 7.* Visual comparisons under W4A6 and W4A4 settings.

*Table 2.* Ablation study of SVD-L on RealSR under W4A6.

| Rank | MUSIQ ↑ | MANIQA ↑ | ClipIQA ↑ | LIQE ↑ | FLOPs (G) | Param (M) |
|------|---------|----------|-----------|--------|-----------|-----------|
| 4 | 66.71 | 0.4432 | 0.5432 | 3.782 | 207.37 | 454.78 |
| 8 | 67.72 | 0.4566 | 0.5671 | 3.980 | 208.22 | 465.39 |
| 16 | 67.80 | 0.4563 | 0.5447 | 3.880 | 209.91 | 486.62 |
| 32 | 68.01 | 0.4560 | 0.5626 | 3.924 | 213.28 | 529.06 |

*Table 3.* Ablation study of VaSMP on RealSR under W4A6.

| Methods | MUSIQ ↑ | MANIQA ↑ | CLIP-IQA ↑ | LIQE ↑ |
|---------|---------|----------|------------|--------|
| Baseline | 65.84 | 0.4286 | 0.5265 | 3.696 |
| +H-SVD | 67.46 | 0.4478 | 0.5475 | 3.834 |
| +H-SVD +MP | 66.84 | 0.4388 | 0.5335 | 3.838 |
| +H-SVD +VaSMP | 67.72 | 0.4566 | 0.5671 | 3.980 |

and FlatQuant (Sun et al., 2025), DiT-specific approaches Q-DiT (Chen et al., 2025) and PTQ4DiT (Wu et al., 2024a), the diffusion-model method Q-Diffusion (Li et al., 2023a), and the image SR-oriented method PassionSR (Zhu et al., 2025). We also include PEFT methods QueST (Wang et al., 2025) and EfficientDM (He et al., 2024).

### 4.2. Main Results

**Quantitative Results.** Tables 1(a) and (b) report quantitative comparisons with representative quantization methods on four real-world SR benchmarks under W4A6 and W4A4, respectively. Under the W4A6 setting, we disable VaTMP since the activation precision is sufficiently high. Even without temporal mixed precision, Q-DiT4SR achieves very close performance compared to the FP model across all datasets and metrics, while consistently outperforming prior approaches. Notably, unlike most other methods, Q-DiT4SR requires no calibration data to reach this performance level.

Under the more aggressive W4A4 setting, we further enable VaTMP to mitigate the accumulation of activation quantization error along the diffusion trajectory. As shown in Table 1(b), Q-DiT4SR achieves the best overall performance, whereas existing methods exhibit consistent degradation across datasets and metrics, indicating limited robustness under low activation precision. This highlights that W4A4 constitutes a more challenging regime for DiT backbones, where adaptive activation precision becomes critical.

*Table 4.* Ablation study of VaTMP on RealSR under W4A4.

| Methods | MUSIQ ↑ | MANIQA ↑ | CLIP-IQA ↑ | LIQE ↑ |
|---|---|---|---|---|
| Baseline | 64.94 | 0.4111 | 0.4899 | 3.191 |
| +H-SVD +VaSMP | 65.83 | 0.4227 | 0.4922 | 3.091 |
| +H-SVD +VaSMP + VaTMP | 66.36 | 0.4367 | 0.4956 | 3.179 |

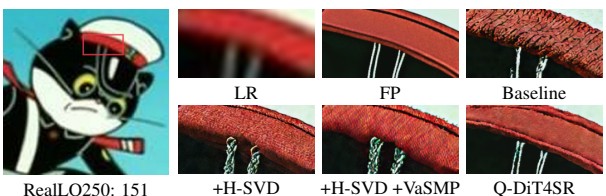

LR    FP    Baseline

RealLQ250: 151   +H-SVD   +H-SVD +VaSMP   Q-DiT4SR

*Figure 8.* Ablation study of H-SVD, VaSMP and VaTMP. Visual comparison (×4) on RealLQ250 under the W4A4 setting.

We further observe that several methods with visibly noisy reconstructions (*e.g.* Q-Diffusion and EfficientDM) may still obtain relatively high scores under certain no-reference IQA metrics such as MANIQA. This observation, also reported by PassionSR, suggests a mismatch between perceptual quality and metric responses in heavily quantized diffusion models, motivating future work on more quantization-aware evaluation criteria for diffusion-based SR.

**Visual Comparison.** Figure 7 presents qualitative comparisons on multiple real-world benchmarks under different quantization settings. The first two rows correspond to the W4A6 configuration, while the last two rows are evaluated under the more aggressive W4A4 setting. Across all datasets, our method consistently reconstructs sharper structures and more faithful high-frequency details, exhibiting clearer edges and richer textures than existing quantization baselines. Notably, even under the challenging W4A4 regime, the visual quality produced by our method Q-DiT4SR remains highly competitive, with only marginal perceptual differences compared to the FP model.

### 4.3. Ablation Study
**Rank Budget of SVD-L.** We study the impact of the SVD-L rank budget in H-SVD under W4A6. In all experiments, the SVD-G is fixed to rank 32, and only the SVD-L is varied. As shown in Table 2, increasing the rank budget from 4 to 8 yields the largest performance gain, while further increasing to 16 or 32 brings marginal or even negative improvements. Meanwhile, both FLOPs and parameter counts increase monotonically with the rank budget, indicating a clear quality–efficiency trade-off. Therefore, we set the rank budget of SVD-L to 8 in all subsequent experiments.

**VaSMP.** We evaluate VaSMP under the W4A6 setting on RealSR, with results summarized in Table 3. Starting from the baseline, incorporating H-SVD yields consistent improvements across all metrics. A naive mixed-precision baseline (MP), which allocates weight bit-widths by minimizing a global end-to-end MSE objective, provides only limited gains and may even degrade certain metrics, suggesting that a purely global criterion fails to capture inter-layer sensi-

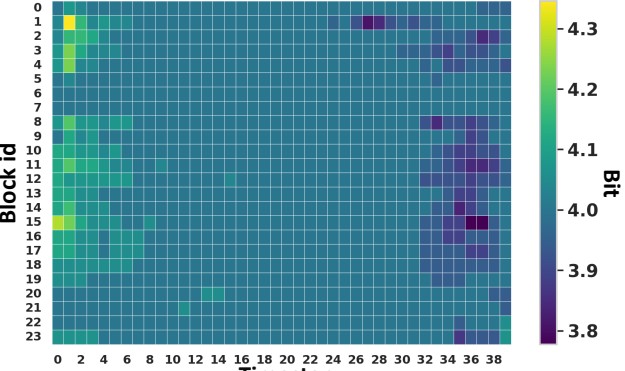

*Figure 9.* VaTMP scheduling result. Heatmap of the average activation bit-width across DiT blocks and diffusion timesteps.

*Table 5.* Practical memory and end-to-end speedup under the W4A4 setting. Peak memory is measured during inference. Speedup is measured relative to the FP model.

| Method | Precision | Peak Mem. (MiB) ↓ | E2E Speedup ↑ |
|---|---|---|---|
| FP | W32A32 | 15085.99 | 1.0× |
| SVDQuant | W4A4 | 3722.83 | ∼4.8× |
| Q-DiT4SR | W4A4 | 3974.64 | ∼4.5× |

tivity differences. In contrast, VaSMP performs cross-layer weight precision allocation guided by weight-variance statistics, explicitly modeling inter-layer variance heterogeneity. This strategy consistently outperforms both H-SVD and MP, achieving the best results across all four IQA metrics.

**VaTMP.** We evaluate VaTMP under the aggressive W4A4 setting on RealSR, with results reported in Table 4. Starting from the W4A4 baseline, combining H-SVD with VaSMP already improves perceptual quality, showing that stronger weight reconstruction and variance-aware weight precision allocation remain effective even under low activation precision. Building on this, VaTMP performs variance-guided temporal activation bit-width allocation across diffusion timesteps, adapting precision to temporal sensitivity. This temporal scheduling consistently brings further gains over H-SVD and VaSMP, achieving the best overall performance. Visual comparisons in Fig. 8 further demonstrate that VaTMP better preserves fine-grained textures and local structures under the same W4A4 budget. The resulting VaTMP scheduling pattern is visualized in Fig. 9.

### 4.4. Practical Runtime and Memory
We profile runtime and memory under W4A4 on a single RTX-4090 48GB GPU with batch size 16. As shown in Table 5, Q-DiT4SR reduces peak memory from 15085.99 MiB to 3974.64 MiB and achieves ∼4.5× end-to-end speedup. Compared with SVDQuant, Q-DiT4SR uses slightly more memory due to the additional H-SVD branch, but maintains comparable acceleration with better reconstruction quality.

Table 6 further breaks down the runtime. While the theoretical operation reduction is 6.14×, the measured end-to-end

*Table 6.* Runtime breakdown of FP and Q-DiT4SR.

| Component | FP (ms) | Q-DiT4SR (ms) | Speedup ↑ |
|---|---|---|---|
| Quantized linear layers | 1580.91 | 175.88 | 8.99× |
| Nonlinear / unquantized ops | 1518.39 | 510.62 | 2.97× |
| Total | 3099.30 | 686.50 | 4.51× |

*Table 7.* Generalization to DreamClear on RealSR.

| Precision | LPIPS ↓ | MUSIQ ↑ | MANIQA ↑ | CLIPIQA ↑ | LIQE ↑ |
|---|---|---|---|---|---|
| FP | 0.3249 | 57.77 | 0.4267 | 0.4695 | 3.082 |
| W4A6 | 0.3691 | 62.94 | 0.5060 | 0.5272 | 3.472 |
| W4A4 | 0.3752 | 56.33 | 0.4005 | 0.4308 | 2.715 |

*Table 8.* Calibration robustness of VaTMP.

| Calibration Set | MUSIQ ↑ | MANIQA ↑ | CLIPIQA ↑ | LIQE ↑ |
|---|---|---|---|---|
| RealSR | 66.36 | 0.4367 | 0.4956 | 3.179 |
| DrealSR | 66.24 | 0.4301 | 0.4963 | 3.296 |
| DIV2K | 65.92 | 0.4228 | 0.4942 | 3.035 |

*Table 9.* Artifact analysis on local regions under W4A4.

| Method | ROI-PSNR ↑ | Edge MAE ↓ | Color MAE ↓ |
|---|---|---|---|
| Baseline | 19.6684 | 0.2757 | 3.7694 |
| H-SVD | 19.8906 | 0.2739 | 3.5504 |
| H-SVD + VaSMP | 19.7209 | 0.2678 | 3.8786 |
| Q-DiT4SR | **20.9590** | **0.2235** | **3.2286** |

speedup is 4.51×, mainly because nonlinear and unquantized components become the dominant bottleneck after linear layers are quantized. The H-SVD branch introduces only about 3% additional latency. Specifically, the global low-rank branch (SVD-G) is executed in parallel with the main quantized branch, following the branch-parallel execution strategy used in Nunchaku (Li et al., 2024). The local block-wise rank-1 branch (SVD-L) is implemented with an efficient Triton kernel (Chen et al., 2021b;a; Dao et al., 2022), inspired by the structured computation pattern of Monarch matrices, which avoids materializing dense local reconstruction matrices. As a result, H-SVD preserves local details while incurring only minimal runtime overhead.

### 4.5. Generalization and Calibration Robustness

To examine backbone generalization, we further evaluate Q-DiT4SR on DreamClear (Ai et al., 2024), another DiT-based Real-ISR architecture. We directly apply the same quantization framework without backbone-specific modification. As shown in Table 7, Q-DiT4SR achieves competitive performance under W4A6 and remains reasonably robust under W4A4, suggesting that H-SVD and variance-aware mixed precision are not restricted to DiT4SR. We interpret the W4A6 improvements over FP cautiously: moderate quantization may act as a mild perceptual regularizer by suppressing unstable high-frequency details, while also exposing the limitation of existing no-reference IQA metrics for quantized generative restoration.

We also examine VaTMP calibration robustness. While the main experiments use 32 LR images from RealSR, we replace them with 32 images from DrealSR and DIV2K (Agustsson & Timofte, 2017), and evaluate on RealSR under W4A4. As shown in Table 8, the performance remains stable across calibration domains, indicating that VaTMP mainly relies on general timestep-wise activation variance patterns rather than dataset-specific statistics.

### 4.6. Artifact Analysis and Limitations

We further analyze W4A4 artifacts, such as color shifts, broken edges, and unstable textures, which can be amplified across diffusion timesteps in quantized generative restoration. To quantify such local degradation, we conduct an ROI study on 32 randomly selected RealSR images by cropping

one $64 \times 64$ region with thin structures, repetitive patterns, or high-contrast details from each image. We report ROI-PSNR for local fidelity, Edge MAE for gradient-domain distortion, and Color MAE for chromatic deviation. As shown in Table 9, H-SVD improves local fidelity and color consistency, while VaSMP further reduces edge distortion; the full Q-DiT4SR with VaTMP achieves the best results on all three metrics. These observations suggest that W4A4 artifacts mainly arise from two error sources in quantized linear layers: weight approximation error and timestep-dependent activation quantization error. H-SVD and VaSMP reduce the former through better weight reconstruction and cross-layer bit allocation, while VaTMP mitigates the latter by assigning higher activation precision to sensitive timesteps.

Despite these improvements, Q-DiT4SR may still produce artifacts on extremely thin, repetitive, or high-contrast textures under W4A4, where small chromatic or structural errors can be visually amplified. Moreover, our current implementation does not use native arbitrary-precision kernels such as W3A3 or W5A5; VaSMP and VaTMP are instead mapped to standard low-bit linear computations. Thus, the practical speedup should be interpreted together with existing kernel support, and integrating our precision schedules with native arbitrary-precision kernels remains future work.

## 5. Conclusion

We propose Q-DiT4SR, the first post-training quantization (PTQ) framework tailored for efficient deployment of Diffusion Transformers (DiTs) in Real-World Image Super-Resolution (Real-ISR). To preserve fine-grained structures under low-bit precision, we propose H-SVD, a hierarchical weight reconstruction strategy that integrates a global low-rank branch with a local rank-1 branch under a matched parameter budget. We further introduce variance-aware spatio-temporal mixed precision, where VaSMP allocates weight bit-widths across layers in a data-free manner, and VaTMP schedules activation precision across diffusion timesteps. Extensive experiments on multiple real-world benchmarks demonstrate that Q-DiT4SR achieves SOTA performance under both W4A6 and W4A4 settings. We hope this work will facilitate efficient deployment of DiT backbones in resource-constrained Real-ISR applications.

## Acknowledgments

This work is supported by the National Natural Science Foundation of China (62501386, 625B1024), CCF-Tencent Rhino-Bird Open Research Fund, and CAAI-Tencent Rhino-Bird Open Research Fund. This work is also sponsored by AI Hundred Schools Program and is carried out using the Ascend AI technology stack.

## Impact Statement

This paper presents work whose goal is to advance the field of Machine Learning. There are many potential societal consequences of our work, none which we feel must be specifically highlighted here.

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
