# Q-DiT4SR: Exploration of Detail-Preserving Diffusion Transformer Quantization for Real-World Image Super-Resolution

## A. H-SVD

### A.1. Supplementary Details

This section supplements the main paper with additional derivation and implementation details of **H-SVD** (Hierarchical SVD), our weight-side reconstruction strategy for preserving fine-grained residual structures under low-bit quantization. We describe its full construction in the Hadamard domain (Yang et al., 2025), the blockwise rank-1 formulation, and the budget-aware design rationale, based on classical low-rank approximation theory (Li et al., 2024).

#### A.1.1. PROBLEM SETUP IN THE HADAMARD DOMAIN

Consider a linear layer with weight matrix $\mathbf{W} \in \mathbb{R}^{out \times in}$. We apply a normalized Hadamard transform along the input dimension:

$$\mathbf{W}_{\mathrm{H}} \triangleq \mathbf{W}\mathbf{H}_n, \tag{1}$$

and perform quantization in this transformed space. A standard decomposition first extracts a full-precision global low-rank approximation and quantizes the remaining residual:

$$\mathbf{W}_{\mathrm{H}} = \mathbf{W}_{\mathrm{SVD\text{-}G}} + \mathbf{W}_{\mathrm{res}},$$
$$\mathbf{W}_{\mathrm{res}} \triangleq \mathbf{W}_{\mathrm{H}} - \mathbf{W}_{\mathrm{SVD\text{-}G}}. \tag{2}$$

While $\mathbf{W}_{\mathrm{SVD\text{-}G}}$ captures global correlations, the residual $\mathbf{W}_{\mathrm{res}}$ often contains spatially heterogeneous and high-frequency patterns. Directly quantizing $\mathbf{W}_{\mathrm{res}}$ is therefore prone to perceptual degradation at low precision, as observed in prior quantization studies (Li et al., 2021; Nagel et al., 2019). H-SVD addresses this issue by introducing an additional *localized* full-precision branch that approximates $\mathbf{W}_{\mathrm{res}}$ under a matched parameter budget, reducing the burden of the subsequent residual quantization.

#### A.1.2. GLOBAL SVD BRANCH

Let the truncated SVD of $\mathbf{W}_{\mathrm{H}}$ be

$$\mathbf{W}_{\mathrm{H}} \approx \mathbf{U}_r \mathbf{\Sigma}_r \mathbf{V}_r^\top \triangleq \mathbf{W}_{\mathrm{SVD\text{-}G}}, \tag{3}$$

[1] Anonymous Institution, Anonymous City, Anonymous Region, Anonymous Country. Correspondence to: Anonymous Author <anon.email@domain.com>.

Preliminary work. Under review by the International Conference on Machine Learning (ICML). Do not distribute.

where $r$ is the target rank (fixed to $r{=}32$ in all experiments). For efficient inference, we use a factored form

$$\mathbf{W}_{\mathrm{SVD\text{-}G}} = \mathbf{AB}, \qquad \mathbf{A} \in \mathbb{R}^{out \times r}, \ \mathbf{B} \in \mathbb{R}^{r \times in}, \tag{4}$$

with the corresponding parameter budget

$$P_{\mathrm{SVD\text{-}G}}(r) = r\,(out + in). \tag{5}$$

#### A.1.3. LOCALIZED SVD BRANCH: BLOCKWISE RANK-1 RESIDUAL APPROXIMATION

**Block partition.** We partition the residual $\mathbf{W}_{\mathrm{res}} \in \mathbb{R}^{out \times in}$ into non-overlapping blocks of size $s_o \times s_i$ (output-by-input), where $s_o \mid out$ and $s_i \mid in$. Let $P \triangleq out/s_o$ and $Q \triangleq in/s_i$ denote the number of blocks along each axis. The $(p, q)$-th block is

$$\mathbf{W}^{(p,q)} \triangleq \mathbf{W}_{\mathrm{res}}[\,ps_o : (p{+}1)s_o, \ qs_i : (q{+}1)s_i\,] \in \mathbb{R}^{s_o \times s_i}, \tag{6}$$

for $p \in \{0, \ldots, P{-}1\}$ and $q \in \{0, \ldots, Q{-}1\}$. The total number of blocks is

$$B = P \cdot Q = \frac{out}{s_o} \cdot \frac{in}{s_i}. \tag{7}$$

**Why rank-1 per block.** A key design choice of H-SVD is to use rank-1 approximation per local block, inspired by efficient low-rank compression techniques (Hu et al., 2022). For a block $\mathbf{W}^{(p,q)}$, its rank-1 approximation is

$$\hat{\mathbf{W}}^{(p,q)} = \sigma_{p,q}\,\mathbf{u}_{p,q}\mathbf{v}_{p,q}^\top,$$
$$\mathbf{u}_{p,q} \in \mathbb{R}^{s_o}, \ \mathbf{v}_{p,q} \in \mathbb{R}^{s_i}, \ \sigma_{p,q} \in \mathbb{R}_+. \tag{8}$$

This form is the optimal rank-1 solution under the Frobenius norm, obtained by keeping the top singular components of $\mathbf{W}^{(p,q)}$. In practice, the residual matrix often exhibits *locally coherent* structures after the global low-rank component is removed; a rank-1 factorization is sufficient to preserve the dominant local direction while keeping the branch extremely lightweight.

**Assembling the localized branch.** All blockwise approximations are assembled into a full matrix $\mathbf{W}_{\mathrm{SVD\text{-}L}} \in \mathbb{R}^{out \times in}$:

$$\mathbf{W}_{\mathrm{SVD\text{-}L}} = \mathrm{Assemble}\Big(\{\hat{\mathbf{W}}^{(p,q)}\}_{p,q}\Big). \tag{9}$$

Since blocks are non-overlapping, the approximation error decomposes additively:

$$\|\mathbf{W}_{\mathrm{res}} - \mathbf{W}_{\mathrm{SVD\text{-}L}}\|_F^2 = \sum_{p,q} \|\mathbf{W}^{(p,q)} - \hat{\mathbf{W}}^{(p,q)}\|_F^2, \tag{10}$$

which provides a clean interpretation: H-SVD improves reconstruction by reducing each local residual block error independently.

### A.1.4. BUDGET MATCHING AND BLOCK-SIZE SELECTION

**Localized branch parameter count.** Each rank-1 block in Eq. (8) stores $(s_o + s_i + 1)$ parameters:

$$P_{\text{blk}}(s_o, s_i) = s_o + s_i + 1. \tag{11}$$

Thus the total parameter budget of the localized branch is

$$P_{\text{SVD-L}}(s_o, s_i) = \frac{out}{s_o} \cdot \frac{in}{s_i} \cdot (s_o + s_i + 1). \tag{12}$$

**Matched-budget constraint.** To ensure that the localized branch does not introduce additional full-precision overhead beyond a standard global SVD reference, we enforce

$$P_{\text{SVD-L}}(s_o, s_i) \lesssim P_{\text{SVD-G}}(r) = r\,(out + in). \tag{13}$$

This constraint directly couples the block granularity to the layer shape. Smaller blocks increase locality but also increase the number of blocks, while larger blocks reduce the number of blocks but may underfit local residual structures. By searching feasible pairs

$$\mathcal{S} = \{(s_o, s_i)\ :\ s_o \mid out,\ s_i \mid in\}, \tag{14}$$

and selecting $(s_o, s_i) \in \mathcal{S}$ satisfying Eq. (13), we obtain a budget-matched localized branch that balances locality and parameter efficiency.

### A.1.5. FINAL RECONSTRUCTION AND QUANTIZED FORWARD WITH H-SVD

**Residual refinement.** After constructing $\mathbf{W}_{\text{SVD-L}}$, the remaining residual to be quantized is

$$\tilde{\mathbf{W}}_{\text{res}} \triangleq \mathbf{W}_{\text{res}} - \mathbf{W}_{\text{SVD-L}}. \tag{15}$$

The final reconstructed quantized weight is then

$$\hat{\mathbf{W}} = \left(\mathbf{W}_{\text{SVD-G}} + \mathbf{W}_{\text{SVD-L}} + Q_w(\tilde{\mathbf{W}}_{\text{res}})\right) \mathbf{H}_n^\top. \tag{16}$$

Compared to directly quantizing $\mathbf{W}_{\text{res}}$, H-SVD removes a structured portion of the residual in full precision, leaving a smaller and less structured $\tilde{\mathbf{W}}_{\text{res}}$ that is easier to compress.

**Forward computation.** Let $\mathbf{x} \in \mathbb{R}^{in}$ be the input activation vector and $Q_G(\mathbf{x})$ denote its Hadamard-domain quantization. The quantized forward pass is given by

$$\begin{aligned}
\mathbf{y} = {}& Q_w(\tilde{\mathbf{W}}_{\text{res}})\, Q_G(\mathbf{x}) \\
& + (\mathbf{W}_{\text{SVD-G}} + \mathbf{W}_{\text{SVD-L}})\, \mathbf{H}_n^\top \mathbf{x}.
\end{aligned} \tag{17}$$

This form makes the intended behavior explicit: most signal energy is preserved through the two full-precision branches (global + local), while the quantized branch only models the reduced residual.

### A.1.6. IMPLEMENTATION NOTES: EFFICIENT APPLICATION OF THE LOCALIZED BRANCH

**Fast rank-1 block multiplication.** For an input vector $\mathbf{x}$, we split it into $Q$ contiguous blocks $\mathbf{x}^{(q)} \in \mathbb{R}^{s_i}$ matching the input partition. For block $(p, q)$, the rank-1 multiplication is

$$\hat{\mathbf{W}}^{(p,q)} \mathbf{x}^{(q)} = \sigma_{p,q}\, \mathbf{u}_{p,q}\big(\mathbf{v}_{p,q}^\top \mathbf{x}^{(q)}\big), \tag{18}$$

which requires one inner product of length $s_i$ and one scaled vector add of length $s_o$. Summing over all $(p, q)$ yields the localized output. This structure is naturally compatible with batched implementations and introduces limited overhead while significantly improving residual preservation under aggressive quantization.

## A.2. Branch-wise Visualization of H-SVD: Roles of SVD-G and SVD-L

To provide an intuitive understanding of how the two full-precision branches in **H-SVD** contribute to restoration quality, we conduct a controlled visualization study on a single full-precision layer `blocks_23_ff_net_2`. Starting from the FP weights of this layer, we explicitly construct three *additive* components in the Hadamard domain: (i) the **global SVD branch** (**SVD-G**), (ii) the **localized SVD branch** (**SVD-L**), and (iii) the **remaining branch** (the leftover residual not covered by the two FP branches). Importantly, to isolate branch functionality, we *do not apply any quantization* in this diagnostic experiment; we only enable/disable branches and inspect the resulting SR outputs.

**Branch configurations and difference convention.** We evaluate the following four forward configurations:

- **Remaining only**: the output produced solely by the remaining branch (without SVD-G or SVD-L);

- **Remaining + SVD-L**: adding the localized SVD branch on top of the remaining branch;

- **Remaining + SVD-G**: adding the global SVD branch on top of the remaining branch;

- **Remaining + SVD-G + SVD-L (H-SVD)**: enabling all three branches simultaneously.

We further visualize *difference maps* by subtracting two restored outputs and converting the magnitude to heatmaps for clearer inspection of structural changes. In the saved filenames, the symbol "−" denotes subtraction in the form *image(A) minus image(B)*. Accordingly, the three key attributions shown in Fig. 1 are:

$$\begin{aligned}
& \underbrace{\text{H-SVD} - \text{Remaining}}_{\text{SVD-G + SVD-L}} \\
& \underbrace{\text{H-SVD} - (\text{Remaining} + \text{SVD-L})}_{\text{SVD-G only}} \\
& \underbrace{\text{H-SVD} - (\text{Remaining} + \text{SVD-G})}_{\text{SVD-L only}}
\end{aligned} \tag{19}$$

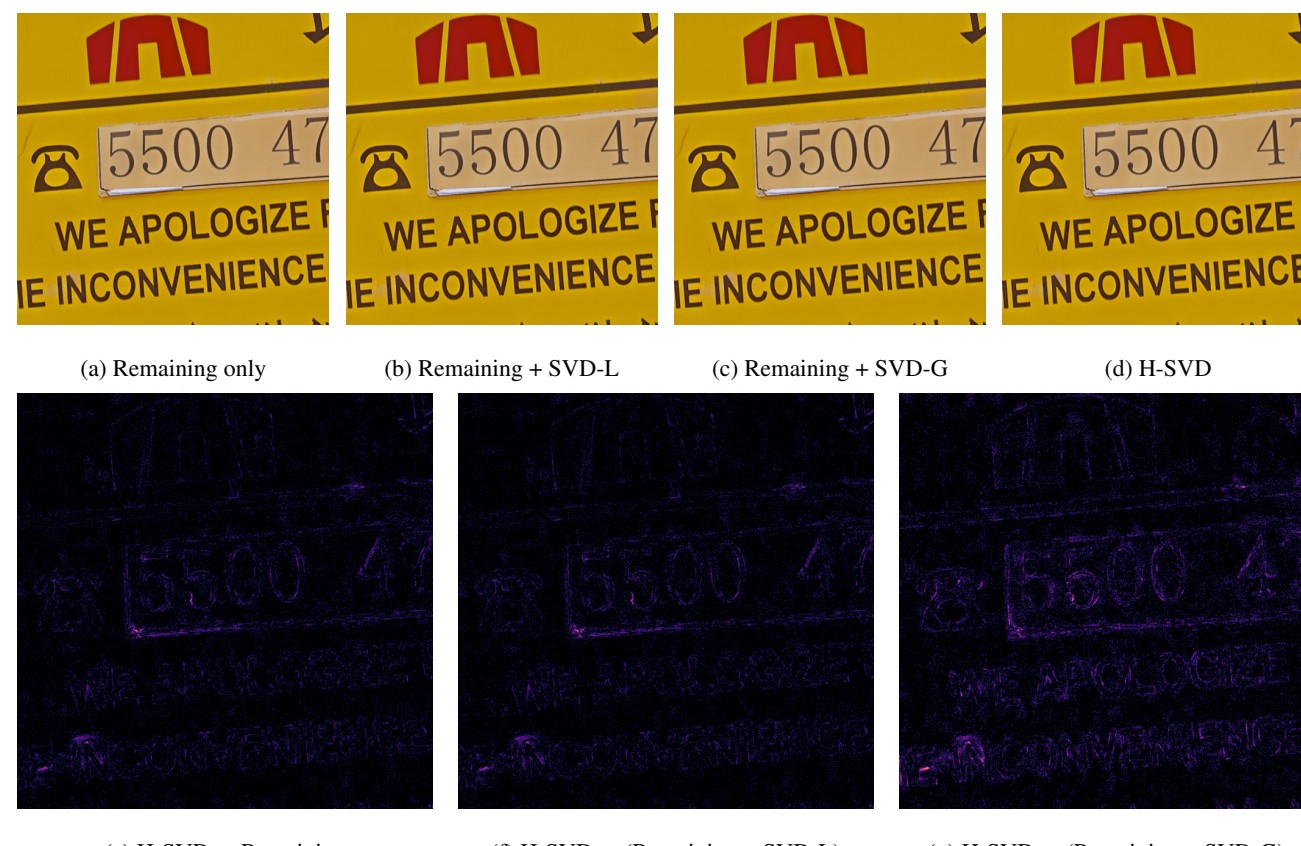

(a) Remaining only     (b) Remaining + SVD-L     (c) Remaining + SVD-G     (d) H-SVD

(e) H-SVD − Remaining     (f) H-SVD − (Remaining + SVD-L)     (g) H-SVD − (Remaining + SVD-G)

*Figure 1.* Branch-wise attribution of H-SVD on `blocks_23_ff_net_2` (no quantization). **Top**: SR outputs obtained by enabling different FP branches. **Bottom**: difference heatmaps computed by subtracting two reconstructions, highlighting spatial regions primarily affected by each branch.

**SVD-G provides a globally-coherent reconstruction backbone.** We first isolate the effect of **SVD-G** through the attribution map H-SVD − (Remaining + SVD-L), shown in Fig. 1(*f*). The response is spatially more *distributed* and aligns with large-scale intensity transitions and major contours (e.g., broad boundaries of the sign and dominant strokes of the digits), indicating that SVD-G captures globally shared components and stabilizes the coarse geometry and overall appearance. This is consistent with the *global low-rank* nature of SVD-G: a shared subspace can efficiently represent dominant correlations across the entire weight matrix, forming a strong global scaffold for reconstruction.

**SVD-L selectively enhances localized high-frequency details.** In contrast, the attribution map H-SVD − (Remaining + SVD-G) in Fig. 1(*g*) reveals the incremental contribution of **SVD-L** when the global branch is already present. Here the activations are significantly more *localized* and concentrate on thin strokes, small typography, and fine boundaries (e.g., sharp digit edges and subtle text structures), which are typical high-frequency cues in real-world SR. This observation matches the design of SVD-L: by allocating blockwise rank-1 capacity across many subregions, SVD-L can represent spatially heterogeneous residual pat-

terns that are difficult to fit with a single global low-rank model, thereby recovering detail-critical structures.

**H-SVD combines complementary global and local behaviors.** Finally, the overall attribution H-SVD − Remaining in Fig. 1(*e*) shows a union of both effects, capturing globally-coherent changes and localized detail refinement. This indicates that SVD-G and SVD-L target *different frequency bands* of layer behavior: SVD-G prioritizes coarse/global structure, while SVD-L focuses on fine/local residuals. Such complementarity explains why H-SVD excels under aggressive compression: it shifts much critical information into full precision, leaving a smaller, less structured remainder that quantizes with less perceptual degradation.

## B. VaSMP

### B.1. Detailed Derivation and Proofs

#### B.1.1. SCALAR UNIFORM QUANTIZATION: HIGH-RATE DISTORTION

We first revisit the classical distortion behavior of uniform quantization under a high-rate approximation. Consider a

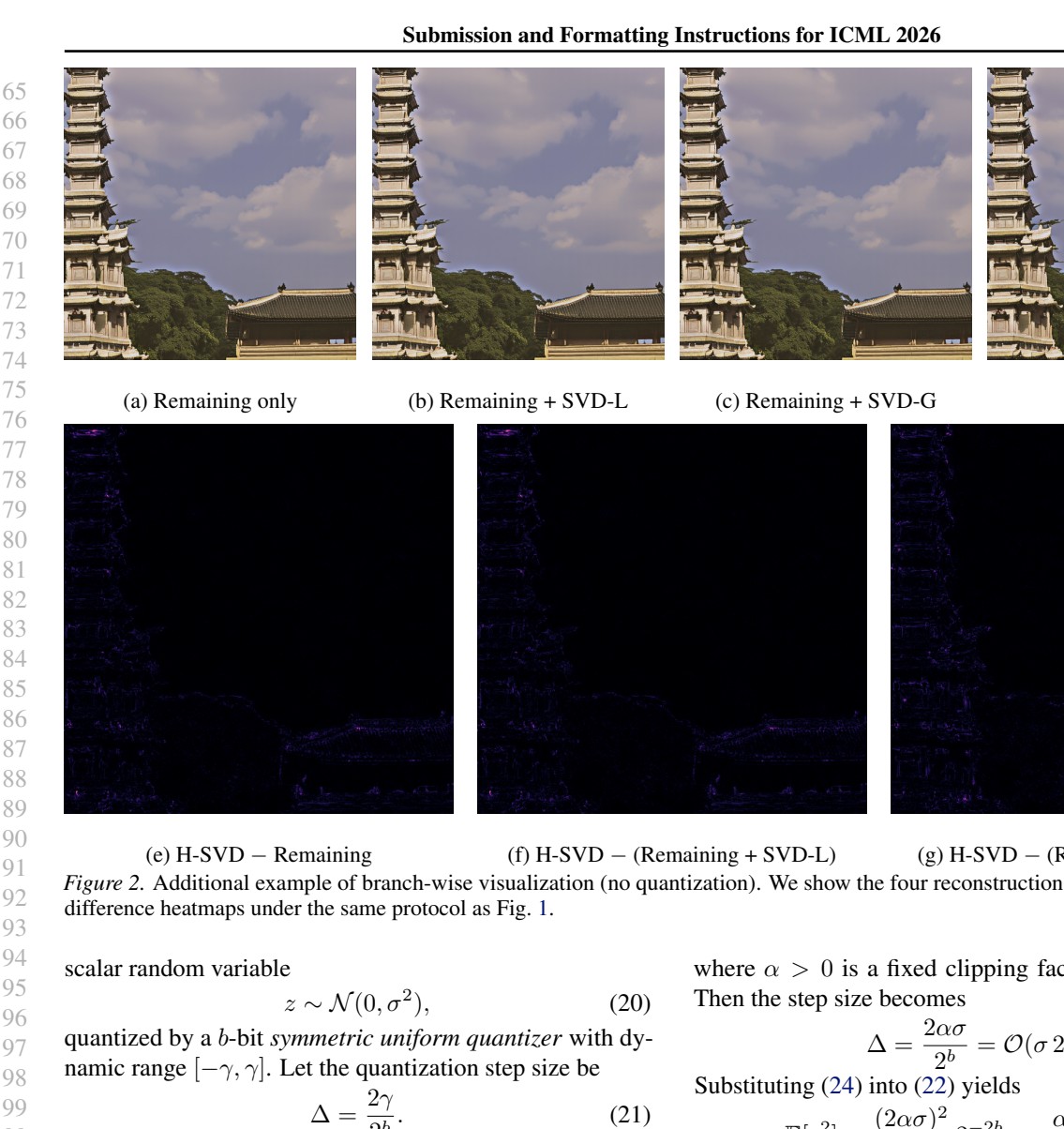

(a) Remaining only       (b) Remaining + SVD-L       (c) Remaining + SVD-G       (d) H-SVD

(e) H-SVD $-$ Remaining       (f) H-SVD $-$ (Remaining + SVD-L)       (g) H-SVD $-$ (Remaining + SVD-G)

*Figure 2.* Additional example of branch-wise visualization (no quantization). We show the four reconstructions and the corresponding difference heatmaps under the same protocol as Fig. 1.

scalar random variable

$$z \sim \mathcal{N}(0, \sigma^2), \tag{20}$$

quantized by a $b$-bit *symmetric uniform quantizer* with dynamic range $[-\gamma, \gamma]$. Let the quantization step size be

$$\Delta = \frac{2\gamma}{2^b}. \tag{21}$$

Denote the quantized value by $\hat{z} = \mathcal{Q}(z)$ and the quantization error by $e = \hat{z} - z$.

**High-rate approximation.** When the quantizer is sufficiently fine (high-rate regime) and the signal rarely saturates, the quantization error can be approximated as being uniformly distributed in $[-\Delta/2, \Delta/2]$ and independent of $z$. Under this standard approximation, the mean squared error (MSE) is

$$\mathbb{E}[e^2] \approx \mathbb{E}\left[\frac{1}{\Delta}\int_{-\Delta/2}^{\Delta/2} u^2\, du\right] = \frac{\Delta^2}{12}. \tag{22}$$

**Step-size scaling for Gaussian clipping.** In practice, $\gamma$ is set proportional to the standard deviation, i.e.,

$$\gamma = \alpha\sigma, \tag{23}$$

where $\alpha > 0$ is a fixed clipping factor (e.g., $\alpha \in [2, 4]$). Then the step size becomes

$$\Delta = \frac{2\alpha\sigma}{2^b} = \mathcal{O}(\sigma\, 2^{-b}). \tag{24}$$

Substituting (24) into (22) yields

$$\mathbb{E}[e^2] \approx \frac{(2\alpha\sigma)^2}{12}\, 2^{-2b} = \underbrace{\frac{\alpha^2}{3}}_{\text{constant}}\, \sigma^2\, 2^{-2b}. \tag{25}$$

Therefore, ignoring constants independent of $b$, we obtain the proportionality:

$$\mathbb{E}[e^2] \propto \sigma^2\, 2^{-2b}. \tag{26}$$

Eq. (25) justifies Eq. (1) in the main text.

### B.1.2. LAYER-WISE QUANTIZATION DISTORTION IN THE HADAMARD DOMAIN

We extend the above result to a weight matrix in layer $\ell$ after the Hadamard transform:

$$\mathbf{W}_{\mathrm{H}}^{(\ell)} \in \mathbb{R}^{out_\ell \times in_\ell}. \tag{27}$$

Let the $o$-th output-channel row be $\mathbf{w}_{\ell,o} \in \mathbb{R}^{1 \times in_\ell}$. We define its empirical variance as

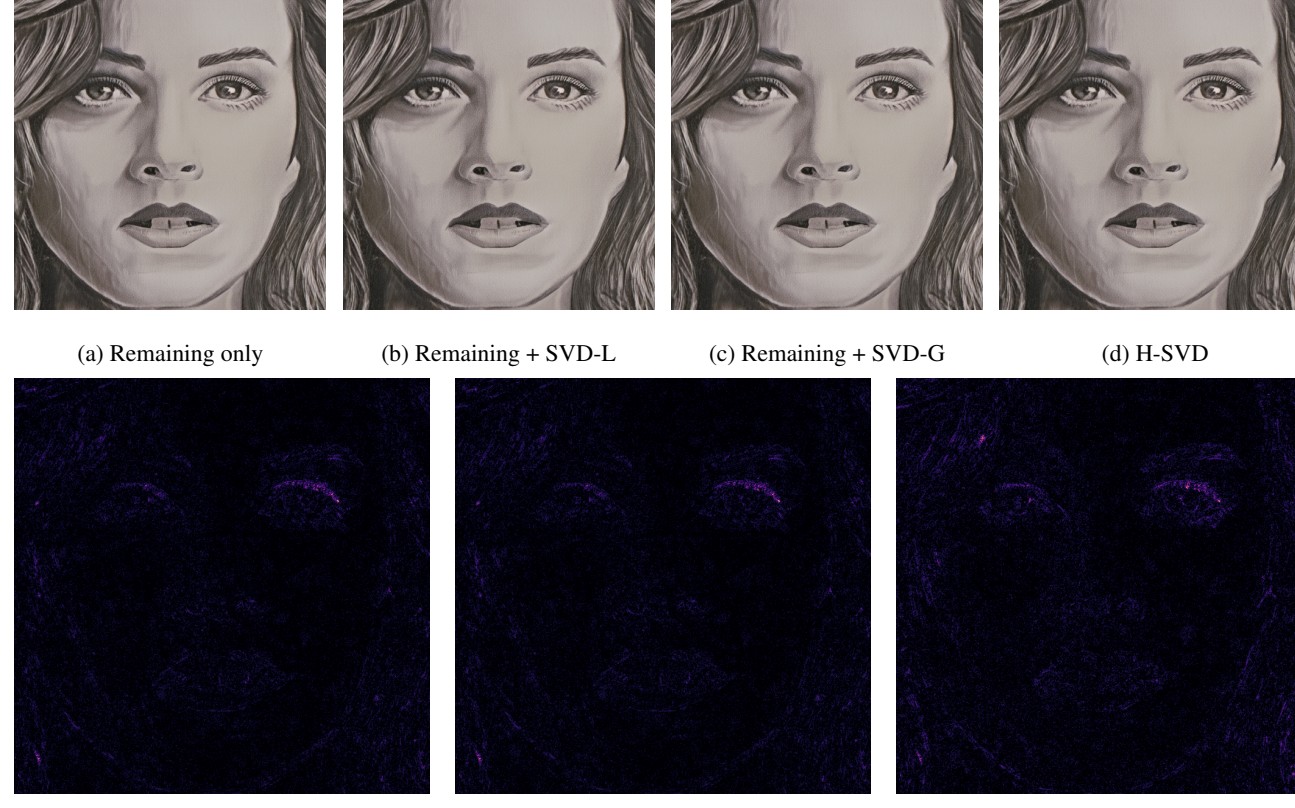

(a) Remaining only     (b) Remaining + SVD-L     (c) Remaining + SVD-G     (d) H-SVD

(e) H-SVD − Remaining     (f) H-SVD − (Remaining + SVD-L)     (g) H-SVD − (Remaining + SVD-G)

*Figure 3.* Another example of branch-wise visualization (no quantization). We include the same set of reconstructions and difference heatmaps for completeness.

$$\sigma_{\ell,o}^2 \triangleq \frac{1}{in_\ell} \sum_{j=1}^{in_\ell} \left(w_{\ell,o,j} - \mu_{\ell,o}\right)^2,$$

$$\mu_{\ell,o} \triangleq \frac{1}{in_\ell} \sum_{j=1}^{in_\ell} w_{\ell,o,j}. \tag{28}$$

We further define the *average output-channel variance*:

$$\bar{\sigma}_\ell^2 \triangleq \frac{1}{out_\ell} \sum_{o=1}^{out_\ell} \sigma_{\ell,o}^2. \tag{29}$$

**Elementwise uniform quantization.** We apply elementwise uniform quantization with a layer bit-width $b_\ell$:

$$\hat{\mathbf{W}}_{\mathrm{H}}^{(\ell)} = \mathbf{Q}\left(\mathbf{W}_{\mathrm{H}}^{(\ell)}; b_\ell\right). \tag{30}$$

Define the distortion as the expected Frobenius reconstruction error:

$$D_\ell(b_\ell) \triangleq \mathbb{E}\left[\left\|\hat{\mathbf{W}}_{\mathrm{H}}^{(\ell)} - \mathbf{W}_{\mathrm{H}}^{(\ell)}\right\|_F^2\right] = \sum_{o=1}^{out_\ell} \sum_{j=1}^{in_\ell} \mathbb{E}\left[e_{\ell,o,j}^2\right], \tag{31}$$

where $e_{\ell,o,j}$ is the quantization error for entry $(o, j)$.

**Row-wise scaling and distortion aggregation.** Assume the quantizer step size for row $o$ is chosen proportional to its standard deviation, i.e., $\Delta_{\ell,o} = \kappa \sigma_{\ell,o} 2^{-b_\ell}$ for some constant $\kappa$ depending on clipping. Then from (25), each entry in row $o$ satisfies

$$\mathbb{E}[e_{\ell,o,j}^2] \propto \sigma_{\ell,o}^2 2^{-2b_\ell}. \tag{32}$$

Summing over all entries in row $o$ gives

$$\sum_{j=1}^{in_\ell} \mathbb{E}[e_{\ell,o,j}^2] \propto in_\ell \sigma_{\ell,o}^2 2^{-2b_\ell}. \tag{33}$$

Finally, summing across all output channels,

$$D_\ell(b_\ell) \propto \sum_{o=1}^{out_\ell} in_\ell \sigma_{\ell,o}^2 2^{-2b_\ell} = in_\ell 2^{-2b_\ell} \sum_{o=1}^{out_\ell} \sigma_{\ell,o}^2$$

$$= (in_\ell \, out_\ell) \left(\frac{1}{out_\ell} \sum_{o=1}^{out_\ell} \sigma_{\ell,o}^2\right) 2^{-2b_\ell}. \tag{34}$$

Let $N_\ell \triangleq in_\ell \, out_\ell$ be the number of parameters in layer $\ell$ (for a linear projection). Using (29), (34) yields the desired form:

$$D_\ell(b_\ell) \propto N_\ell \, \bar{\sigma}_\ell^2 \, 2^{-2b_\ell}. \tag{35}$$

Eq. (35) provides a rigorous justification for Eq. (2) in the main text, highlighting that, under a shared uniform quantization design, the dominant layer sensitivity is captured by the average output-channel variance $\bar{\sigma}_\ell^2$.

### B.1.3. VARIANCE-AWARE CONTINUOUS BIT ALLOCATION: CLOSED-FORM SOLUTION

We now derive the closed-form bit allocation rule used by VaSMP. Let $\mathcal{A}$ denote the set of *active* layers participating in inference:

$$\mathcal{A} \triangleq \{\ell : \text{layer } \ell \text{ participates in the forward computation}\}. \tag{36}$$

Layers outside $\mathcal{A}$ are assigned a fixed bit-width by design.

**Optimization objective.** Motivated by (35), we minimize the surrogate total distortion:

$$\min_{\{b_\ell\}_{\ell \in \mathcal{A}}} \sum_{\ell \in \mathcal{A}} N_\ell \bar{\sigma}_\ell^2 2^{-2b_\ell}, \tag{37}$$

subject to a target *weighted average* bit-width constraint:

$$\sum_{\ell \in \mathcal{A}} w_\ell b_\ell = B_{\text{target}} \sum_{\ell \in \mathcal{A}} w_\ell, \qquad w_\ell = \max(N_\ell, 1). \tag{38}$$

The weights $w_\ell$ ensure that larger layers (in parameter count) contribute proportionally to the bit budget.

**Lagrangian and KKT optimality.** Define the Lagrangian:

$$\mathcal{L}(\{b_\ell\}, \lambda) = \sum_{\ell \in \mathcal{A}} N_\ell \bar{\sigma}_\ell^2 2^{-2b_\ell}$$
$$+ \lambda \left( \sum_{\ell \in \mathcal{A}} w_\ell b_\ell - B_{\text{target}} \sum_{\ell \in \mathcal{A}} w_\ell \right). \tag{39}$$

where $\lambda$ is the multiplier associated with (38).

Taking derivatives and setting stationarity for each $\ell \in \mathcal{A}$:

$$\frac{\partial \mathcal{L}}{\partial b_\ell} = N_\ell \bar{\sigma}_\ell^2 \cdot \frac{\partial}{\partial b_\ell} \left( 2^{-2b_\ell} \right) + \lambda w_\ell = 0, \tag{40}$$

$$\frac{\partial}{\partial b_\ell} \left( 2^{-2b_\ell} \right) = 2^{-2b_\ell} \cdot (-2 \ln 2), \tag{41}$$

hence

$$-2(\ln 2) N_\ell \bar{\sigma}_\ell^2 2^{-2b_\ell} + \lambda w_\ell = 0. \tag{42}$$

Rearranging gives

$$2^{-2b_\ell} = \frac{\lambda w_\ell}{2(\ln 2) N_\ell \bar{\sigma}_\ell^2}. \tag{43}$$

Taking $\log_2(\cdot)$ on both sides:

$$-2b_\ell = \log_2(\lambda) + \log_2(w_\ell) - \log_2\left(2 \ln 2\right)$$
$$- \log_2(N_\ell) - \log_2(\bar{\sigma}_\ell^2). \tag{44}$$

Thus,

$$b_\ell = \frac{1}{2} \log_2(\bar{\sigma}_\ell^2) + \frac{1}{2} \log_2(N_\ell) - \frac{1}{2} \log_2(w_\ell) + c, \tag{45}$$

where $c$ collects all constants independent of $\ell$:

$$c \triangleq -\frac{1}{2} \log_2(\lambda) + \frac{1}{2} \log_2\left(2 \ln 2\right). \tag{46}$$

**Simplification under $w_\ell = N_\ell$.** In VaSMP, we use $w_\ell = \max(N_\ell, 1)$, and for practical layers $N_\ell \geq 1$ holds, hence

$w_\ell = N_\ell$. Then the $N_\ell$-dependent terms in (45) cancel:

$$b_\ell = \frac{1}{2} \log_2(\bar{\sigma}_\ell^2) + c. \tag{47}$$

**Determining the constant $c$ from the budget constraint.** Substitute (47) into (38):

$$\sum_{\ell \in \mathcal{A}} w_\ell b_\ell = \sum_{\ell \in \mathcal{A}} w_\ell \left( \frac{1}{2} \log_2(\bar{\sigma}_\ell^2) + c \right) \tag{48}$$

$$= B_{\text{target}} \sum_{\ell \in \mathcal{A}} w_\ell. \tag{49}$$

Divide both sides by $\sum_{\ell \in \mathcal{A}} w_\ell$:

$$\frac{1}{2} \cdot \frac{\sum_{\ell \in \mathcal{A}} w_\ell \log_2(\bar{\sigma}_\ell^2)}{\sum_{\ell \in \mathcal{A}} w_\ell} + c = B_{\text{target}}. \tag{50}$$

Define the weighted mean

$$\overline{\log_2 \bar{\sigma}} \triangleq \frac{\sum_{\ell \in \mathcal{A}} w_\ell \log_2(\max(\bar{\sigma}_\ell^2, \epsilon))}{\sum_{\ell \in \mathcal{A}} w_\ell}, \tag{51}$$

where $\epsilon > 0$ is a small constant for numerical stability. Then

$$c = B_{\text{target}} - \frac{1}{2} \overline{\log_2 \bar{\sigma}}. \tag{52}$$

Plugging back into (47) yields the closed-form solution:

$$b_\ell^* = B_{\text{target}}$$
$$+ \frac{1}{2} \left( \log_2(\max(\bar{\sigma}_\ell^2, \epsilon)) - \overline{\log_2 \bar{\sigma}} \right), \qquad \ell \in \mathcal{A}. \tag{53}$$

which matches Eq. (6) in the main text.

**Interpretation.** Eq. (53) shows that layers with higher variance $\bar{\sigma}_\ell^2$ receive larger bit-widths, while low-variance layers receive fewer bits, and the allocation is automatically normalized to satisfy the global average target $B_{\text{target}}$.

### B.1.4. GREEDY DISCRETIZATION: DERIVATION OF THE GAIN SCORE

The continuous solution (53) produces real-valued bit-widths. We discretize them into integer bit-widths for deployment under the bounds $b_{\min} \leq b_\ell \leq b_{\max}$.

**Layer distortion as a function of bits.** From (35), the distortion follows

$$D_\ell(b) = C_\ell 2^{-2b}, \qquad C_\ell \triangleq \kappa_\ell N_\ell \bar{\sigma}_\ell^2, \tag{54}$$

where $\kappa_\ell > 0$ aggregates constants from quantizer design. For prioritization, constants independent of $b$ do not affect ordering.

**Marginal distortion reduction from adding one bit.** The reduction in distortion when increasing $b$ by one is

$$\Delta D_\ell(b) \triangleq D_\ell(b) - D_\ell(b+1) = C_\ell 2^{-2b} - C_\ell 2^{-2(b+1)}$$
$$= C_\ell 2^{-2b} \left( 1 - 2^{-2} \right) = \frac{3}{4} C_\ell 2^{-2b}. \tag{55}$$

**Algorithm 1** VaSMP: Variance-Aware Cross-Layer Mixed-Precision Bit Allocation

**Require:** Active layers $\mathcal{A}$; layer statistics $\{(N_\ell, \bar{\sigma}_\ell^2)\}$; target average bit-width $B_{\text{target}}$; integer bounds $b_{\min}, b_{\max}$; stability $\epsilon$.

**Ensure:** Integer bit-widths $\{b_\ell\}_{\ell \in \mathcal{A}}$.

1: Set weights $w_\ell \leftarrow \max(N_\ell, 1)$ for all $\ell \in \mathcal{A}$.
2: Compute weighted mean

$$\overline{\log_2 \bar{\sigma}} \leftarrow \frac{\sum_{\ell \in \mathcal{A}} w_\ell \log_2(\max(\bar{\sigma}_\ell^2, \epsilon))}{\sum_{\ell \in \mathcal{A}} w_\ell}.$$

3: Compute real-valued solution

$$b_\ell^* \leftarrow B_{\text{target}} + \frac{1}{2} \left( \log_2(\max(\bar{\sigma}_\ell^2, \epsilon)) - \overline{\log_2 \bar{\sigma}} \right), \quad \forall \ell \in \mathcal{A}.$$

4: Initialize integers

$$b_\ell \leftarrow \text{clip}(\lfloor b_\ell^* \rceil, b_{\min}, b_{\max}), \quad \forall \ell \in \mathcal{A}.$$

5: Let residual budget be

$$\Delta \leftarrow B_{\text{target}} \sum_{\ell \in \mathcal{A}} w_\ell - \sum_{\ell \in \mathcal{A}} w_\ell b_\ell.$$

6: **while** $\Delta > 0$ **do**
7:   Compute priority for each layer:

$$\text{Gain}_\ell \leftarrow \bar{\sigma}_\ell^2 \, 4^{-b_\ell}, \quad \forall \ell \in \mathcal{A}.$$

8:   Select

$$\ell^\star \leftarrow \arg \max_{\ell \in \mathcal{A}, \, b_\ell < b_{\max}} \text{Gain}_\ell.$$

9:   $b_{\ell^\star} \leftarrow b_{\ell^\star} + 1$.
10:   $\Delta \leftarrow \Delta - w_{\ell^\star}$.
11: **end while**
12: **return** $\{b_\ell\}_{\ell \in \mathcal{A}}$.

Therefore, the *priority score* for allocating one additional bit to layer $\ell$ can be taken as

$$\text{Gain}_\ell(b) \propto C_\ell \, 2^{-2b} \propto N_\ell \, \bar{\sigma}_\ell^2 \, 2^{-2b} = N_\ell \, \bar{\sigma}_\ell^2 \, 4^{-b}. \quad (56)$$

**Gain used in the main text.** In our implementation, the global budget is controlled via parameter-weighted averaging, and we empirically found that the ordering is dominated by $\bar{\sigma}_\ell^2$ and $b_\ell$. Thus, for simplicity and robustness, we use the following layer-wise gain proxy:

$$\text{Gain}_\ell \propto \bar{\sigma}_\ell^2 \, 4^{-b_\ell}, \quad (57)$$

which matches Eq. (7) in the main text (up to a constant factor). Intuitively, layers with high variance and low current precision yield the largest marginal benefit when assigning extra bits.

B.1.5. COMPLETE DISCRETIZATION PROCEDURE

We summarize the discretization procedure used by VaSMP for deployment.

**Remarks.** The above procedure is entirely *data-free*: all quantities $(N_\ell, \bar{\sigma}_\ell^2)$ are computed offline from the transformed weights, without using calibration data or second-order information (e.g., Hessian (Dong et al., 2019; Yao et al., 2021)). Moreover, the greedy refinement guarantees that the final integer solution respects the global bit budget while prioritizing layers with the largest marginal distortion reduction.

**B.2. Weight Bit-Width Allocation Statistics**

This section provides a detailed analysis of the weight bit-width allocation produced by our VaSMP strategy. VaSMP assigns per-layer weight precision from a discrete candidate set $\{2, 3, 4, 5, 6, 7, 8\}$ under a global budget, aiming to preserve perceptual quality while minimizing quantization distortion, following the principles of mixed-precision quantization (Wang et al., 2019; Dong et al., 2019). For completeness, we note that layers that are *not involved in the forward pass* (e.g., unused branches) or exhibit *zero variance* are directly assigned 1-bit quantization as a degenerate case. Such layers contribute negligible information flow and are therefore excluded from the average-bit budget accounting reported in this section, ensuring that the computed mean bit-width faithfully reflects the effective precision allocation on actively-used layers. Under this protocol, the resulting parameter-weighted average bit-width is **4.0009**, which closely matches the target budget of 4 bits, demonstrating that VaSMP achieves accurate global budget control while allowing flexible, variance-aware precision allocation at the layer level.

**Overall allocation distribution.** We first summarize the global bit-width statistics. Fig. 4(*left*) shows that VaSMP allocates 5-bit to 237 layers $(47.84 - bit$ to $152 layers (30.6$ This long-tail distribution indicates VaSMP preserves higher precision for most layers, while aggressively quantizing a small set to meet the overall bit budget.

**Parameter-aware budget utilization.** Beyond layer counts, we analyze the *parameter share* contributed by each precision level. Fig. 4(*middle-left*) shows that 5-bit layers account for 42.2% of total parameters, followed by 3-bit (30.5%), 4-bit (21.6%), and 2-bit (5.8%). Interestingly, despite having fewer layers, the 3-bit group occupies a substantial portion of parameters, suggesting that VaSMP preferentially applies 3-bit to large-matrix layers that are relatively insensitive under our variance-based distortion proxy. In contrast, 4-bit covers more layers but contributes a smaller parameter share, implying that it is mostly assigned to medium/small matrices. Overall, the parameter-weighted average precision is approximately 4.01 bits, which matches the intended global budget and confirms

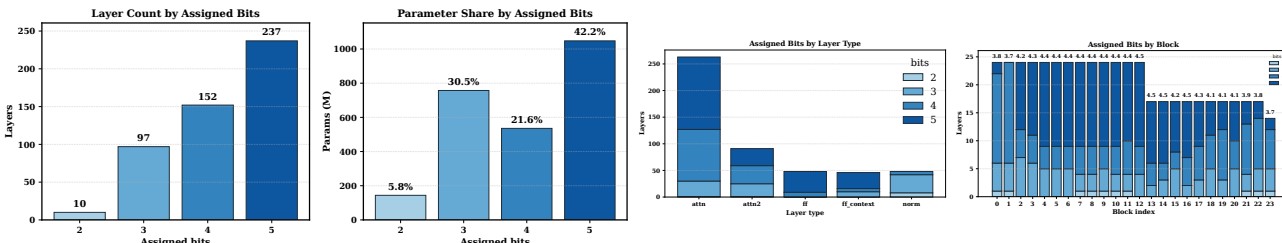

*Figure 4.* VaSMP weight bit-width allocation statistics. **Left:** layer count by assigned bit-width. **Middle-left:** parameter share by assigned bit-width. **Middle-right:** assigned bit-widths grouped by layer type. **Right:** assigned bit-width distribution across Transformer blocks.

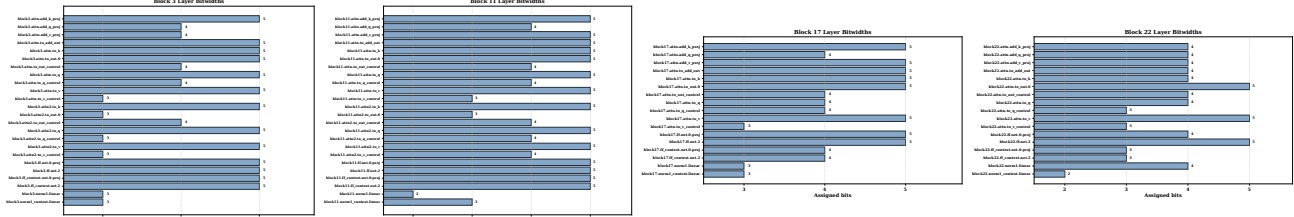

*Figure 5.* Per-layer VaSMP weight bit-width allocation within representative Transformer blocks. VaSMP preserves higher precision for the main attention/MLP projections in earlier and middle blocks, while progressively reducing precision for auxiliary `control`, context, and normalization modules in later blocks.

that VaSMP achieves accurate budget control while keeping precision where it matters most.

**Allocation by module type.** We further break down the assignments by layer type in Fig. 4(*middle-right*). Attention-related projections (`attn` and `attn2`) dominate the model depth and are predominantly allocated 4–5 bits, reflecting their high sensitivity to weight noise (Wu et al., 2024): small perturbations in attention projections can directly alter token mixing patterns and thus affect fine-grained texture reconstruction in DiT-based models (Peebles & Xie, 2023; Duan et al., 2025). Feed-forward components (`ff` and `ff_context`) receive consistently high precision, with a clear bias toward 5-bit, indicating that the MLP-style transformations remain critical for preserving high-frequency details in restoration. In contrast, normalization layers (`norm`) are mostly assigned low precision (2–3 bits), suggesting that their scaling/shift parameters are relatively robust to quantization and can be compressed aggressively with negligible impact on perceptual fidelity. This type-wise trend aligns well with the intuition that VaSMP automatically protects *information-transforming* modules (attention/MLP) while compressing *re-normalization* modules more aggressively.

**Layer-wise evolution across Transformer blocks.** Finally, we visualize the allocation pattern across the 24 Transformer blocks in Fig. 4(*right*). VaSMP yields a clear depth-dependent structure: early blocks (e.g., block 0–1) are assigned lower average precision ($\approx 3.7$–3.8 bits), while the middle stage (block 2–14) stays consistently high ($\approx 4.2$–4.5 bits). Toward the end, the allocated precision gradually decreases (e.g., block 18–23 drops to $\approx 4.1 \rightarrow 3.7$ bits). This pattern suggests that mid-level representations are the most sensitive under low-bit weight quantization, likely be-

cause they are responsible for refining semantic structures into restoration-specific feature bases. By contrast, the earliest blocks mainly perform coarse feature extraction and can tolerate more quantization, while the final blocks contain more normalization and lightweight adaptation that does not require uniformly high weight precision. Overall, the allocation exhibits a principled "protect-the-middle" behavior, providing an interpretable explanation of how VaSMP distributes the limited precision budget across depth.

**Per-block layer-wise bit-width assignments.** To further understand *where* VaSMP spends its limited weight precision budget, we visualize the per-layer bit-width decisions within several representative Transformer blocks in Fig. 5. Across early/middle blocks (e.g., block 3 and block 11), VaSMP consistently assigns high precision (4–5 bits) to the *core* attention projections (`to_k`/`to_q`/`to_v` and `to_out`) and the feed-forward matrices (`ff.*`), while compressing auxiliary `control` branches and normalization linears to lower precision (2–3 bits). This indicates that VaSMP prioritizes preserving the main token-mixing and feature-transformation paths that directly govern structure reconstruction, and treats the modulation components as comparatively robust to weight quantization noise. Interestingly, for block 3/11, several `attn2` modules (e.g., `to_out` and `control` projections) are allocated only 3–4 bits, suggesting lower sensitivity for these secondary attention transformations under our variance-based distortion proxy. As depth increases, VaSMP gradually becomes more aggressive: in block 17, the query-related projections are reduced to 4 bits and the context MLP (`ff_context`) shifts to 4 bits, while the main MLP branch remains 5 bits, reflecting a selective precision drop that preserves the most critical pathways. In the late block 22, most attention projections

are assigned 3–4 bits with only a few remaining at 5 bits (notably `to_out` and `to_v`), and the context branch as well as normalization become even lower (2–3 bits). This depth-dependent pattern complements the global statistics in Fig. 4: VaSMP allocates higher precision to mid-level processing where feature refinement is most sensitive, while exploiting the redundancy of late-stage modulation/normalization to meet the budget with minimal perceptual degradation.

## C. VaTMP

### C.1. Detailed Derivation and Proofs

#### C.1.1. NOTATION AND SETUP

For a diffusion timestep $t$ and layer $\ell$ in a Diffusion Transformer (Peebles & Xie, 2023; Duan et al., 2025), we denote the token activations by $\mathbf{X}^{(\ell,t)} \in \mathbb{R}^{T \times C}$, where $T$ is the number of tokens and $C$ is the channel dimension. Following the preliminaries, we apply the normalized Hadamard transform on the channel axis (Yang et al., 2025):

$$\mathbf{Z}^{(\ell,t)} = \mathbf{X}^{(\ell,t)} \mathbf{H}_n, \qquad \mathbf{H}_n^\top \mathbf{H}_n = \mathbf{I}. \quad (58)$$

We define the timestep-wise sensitivity proxy $v_{\ell,t}$ as the mean token variance of $\mathbf{Z}^{(\ell,t)}$:

$$v_{\ell,t} \triangleq \text{mean\_token\_var}(\ell, t), \quad (59)$$

implemented as the *average channel variance* aggregated over tokens and samples. Concretely, for a minibatch $\{\mathbf{Z}_s^{(\ell,t)}\}_{s=1}^S$, where $\mathbf{Z}_s^{(\ell,t)} \in \mathbb{R}^{T \times C}$, we compute

$$v_{\ell,t} = \frac{1}{S} \sum_{s=1}^S \left( \frac{1}{T} \sum_{\tau=1}^T \text{Var}\left( \mathbf{Z}_s^{(\ell,t)}[\tau, :] \right) \right), \quad (60)$$

where $\text{Var}(\cdot)$ is the variance over the channel dimension.

**Goal.** For each layer $\ell$, VaTMP allocates activation bit-widths $\{b_{\ell,t}\}_{t=1}^{T_\ell}$ across timesteps within the same layer, under a target average activation-bit budget $B_{\text{target}}^{\text{act}}$. The core objective is to minimize a distortion surrogate that is provably proportional to $v_{\ell,t}$ under a Gaussianized assumption.

#### C.1.2. UNIFORM QUANTIZATION WITH CLIPPING

We consider symmetric uniform quantization with clipping. For a scalar input $z \in \mathbb{R}$, clipping bound $A > 0$, and bit-width $b \in \mathbb{N}$, define

$$Q_{A,b}(z) = \Delta \, \text{round}\left( \frac{\text{clip}(z, -A, A)}{\Delta} \right), \quad (61)$$

$$\Delta = \frac{2A}{2^b - 1}.$$

Here, $\text{clip}(z, -A, A) = \min(\max(z, -A), A)$ and $\text{round}(\cdot)$ rounds to the nearest integer.

**Distortion definition.** The per-scalar MSE distortion is

$$d(A, b; z) \triangleq \mathbb{E}\left[ (z - Q_{A,b}(z))^2 \right]. \quad (62)$$

#### C.1.3. SCALING LAW: DISTORTION IS PROPORTIONAL TO VARIANCE

A key property underlying VaTMP is that, for a Gaussian input, the optimal clipped-uniform quantization distortion scales linearly with the input variance.

**Lemma C.1** (Variance scaling of clipped-uniform MSE). *Let $X \sim \mathcal{N}(0, v)$ with $v > 0$, and define $Z \triangleq X/\sqrt{v} \sim \mathcal{N}(0, 1)$. For any $A > 0$ and bit-width $b$, let $a \triangleq A/\sqrt{v}$. Then*

$$\mathbb{E}\left[ (X - Q_{A,b}(X))^2 \right] = v \cdot \mathbb{E}\left[ (Z - Q_{a,b}(Z))^2 \right]. \quad (63)$$

*Proof.* Using $X = \sqrt{v} \, Z$ and $A = \sqrt{v} \, a$, the step size in (61) becomes

$$\Delta = \frac{2A}{2^b - 1} = \sqrt{v} \cdot \frac{2a}{2^b - 1} = \sqrt{v} \, \delta, \, \delta \triangleq \frac{2a}{2^b - 1}. \quad (64)$$

Moreover,

$$\text{clip}(X, -A, A) = \sqrt{v} \, \text{clip}(Z, -a, a). \quad (65)$$

Therefore,

$$\begin{aligned}
Q_{A,b}(X) &= \Delta \, \text{round}\left( \frac{\text{clip}(X, -A, A)}{\Delta} \right) \\
&= \sqrt{v} \, \delta \, \text{round}\left( \frac{\sqrt{v} \, \text{clip}(Z, -a, a)}{\sqrt{v} \, \delta} \right) \\
&= \sqrt{v} \underbrace{\left( \delta \, \text{round}\left( \frac{\text{clip}(Z, -a, a)}{\delta} \right) \right)}_{Q_{a,b}(Z)} \\
&= \sqrt{v} \, Q_{a,b}(Z). \quad (66)
\end{aligned}$$

Thus,

$$X - Q_{A,b}(X) = \sqrt{v} \, (Z - Q_{a,b}(Z)), \quad (67)$$

and taking expectation yields (63). $\square$

**Normalized distortion coefficient.** Lemma C.1 implies that, under $X \sim \mathcal{N}(0, v)$, the quantization MSE admits a separable form: a variance factor $v$ times a bit-dependent coefficient. We therefore define the optimal normalized distortion coefficient for the standard Gaussian:

$$\kappa(b) \triangleq \min_{a > 0} \mathbb{E}_{Z \sim \mathcal{N}(0,1)}\left[ (Z - Q_{a,b}(Z))^2 \right], \quad (68)$$

which depends *only* on the bit-width $b$. In practice, $\kappa(b)$ is precomputed offline for candidate bit-widths $b \in \mathcal{B}$ by 1D search over $a$ (e.g., grid search + local refinement).

**Corollary C.2** (Optimal distortion for $\mathcal{N}(0, v)$). *Let $X \sim \mathcal{N}(0, v)$. Then the minimum MSE over clipping bound $A$ satisfies*

$$\min_{A > 0} \mathbb{E}\left[ (X - Q_{A,b}(X))^2 \right] = v \cdot \kappa(b). \quad (69)$$

*Proof.* Apply Lemma C.1 and substitute $a = A/\sqrt{v}$:

$$\begin{aligned}
\min_{A > 0} \mathbb{E}\left[ (X - Q_{A,b}(X))^2 \right] &= \min_{a > 0} v \cdot \mathbb{E}\left[ (Z - Q_{a,b}(Z))^2 \right] \\
&= v \cdot \kappa(b). \quad (70)
\end{aligned}$$

$\square$

### C.1.4. FROM SCALAR DISTORTION TO ACTIVATION DISTORTION PROXY

We now justify this equation, $D_{\ell,t}(b) \propto \kappa(b)\, v_{\ell,t}$.

**Gaussianized activation assumption.** After Hadamard mixing, token activations in $\mathbf{Z}^{(\ell,t)}$ are empirically close to Gaussian with approximately zero mean (Yang et al., 2025), and exhibit a clear variance trend over timesteps in diffusion models (Ho et al., 2020; Song et al., 2021). We model an activation element at layer $\ell$ and timestep $t$ as

$$Z^{(\ell,t)} \sim \mathcal{N}(0, v_{\ell,t}), \tag{71}$$

where $v_{\ell,t}$ is measured by (60).

**Distortion proxy.** By Corollary C.2, the minimum achievable per-scalar distortion under bit-width $b$ satisfies

$$\min_{A>0} \mathbb{E}\left[(Z^{(\ell,t)} - Q_{A,b}(Z^{(\ell,t)}))^2\right] = v_{\ell,t}\, \kappa(b). \tag{72}$$

Summing across activation elements introduces only a multiplicative factor (independent of $b$) under the assumption that the element-wise variance is dominated by the layer-timestep statistic $v_{\ell,t}$. Therefore, a valid surrogate for comparing candidate bit-widths at $(\ell, t)$ is

$$D_{\ell,t}(b) \propto \kappa(b)\, v_{\ell,t}. \tag{73}$$

**Remarks.** Eq. (73) explains why timestep-wise variance serves as a reliable *intra-layer* sensitivity indicator: for a fixed bit-width $b$, higher $v_{\ell,t}$ necessarily implies larger expected distortion, hence merits higher precision.

### C.1.5. TEMPORAL SCHEDULING AS A RESOURCE-CONSTRAINED SEGMENTATION PROBLEM

For each layer $\ell$, we adopt a piecewise-constant schedule across timesteps. Let $T_\ell$ be the number of diffusion timesteps (typically shared across layers). We assign timestep-wise integer bits $b_{\ell,t} \in \mathcal{B}$ under an unweighted average-bit budget:

$$\sum_{t=1}^{T_\ell} b_{\ell,t} \le B_\ell, \qquad B_\ell \triangleq \left\lfloor B_{\text{target}}^{\text{act}}\, T_\ell \right\rfloor. \tag{74}$$

The total surrogate distortion is

$$\min_{\{b_{\ell,t} \in \mathcal{B}\}} \sum_{t=1}^{T_\ell} \kappa(b_{\ell,t})\, v_{\ell,t} \quad \text{s.t.} \quad \sum_{t=1}^{T_\ell} b_{\ell,t} \le B_\ell. \tag{75}$$

**Piecewise-constant constraint.** We further restrict the solution to be constant on contiguous segments. Let a segmentation be defined by boundaries

$$1 = t_0 < t_1 < \cdots < t_M = T_\ell + 1, \tag{76}$$

and assign a bit-width $b_m \in \mathcal{B}$ to each segment $[t_{m-1}, t_m)$. Then

$$b_{\ell,t} = b_m, \qquad \forall t \in [t_{m-1}, t_m). \tag{77}$$

The segment cost becomes

$$\text{SegCost}_\ell(t_{m-1}, t_m; b_m) = \kappa(b_m) \sum_{t=t_{m-1}}^{t_m-1} v_{\ell,t}. \tag{78}$$

The segment consumes a budget of $(t_m - t_{m-1}) \cdot b_m$ bits. Therefore, the segmentation problem is

$$\min_{M, \{t_m\}, \{b_m \in \mathcal{B}\}} \sum_{m=1}^{M} \kappa(b_m) \sum_{t=t_{m-1}}^{t_m-1} v_{\ell,t}$$
$$\text{s.t.} \quad \sum_{m=1}^{M} (t_m - t_{m-1})\, b_m \le B_\ell. \tag{79}$$

This is a resource-constrained shortest-path problem over a DAG of segments.

### C.1.6. DYNAMIC PROGRAMMING SOLUTION AND OPTIMALITY

We provide a rigorous DP formulation that yields the globally optimal segmented schedule under the objective.

**Prefix sums for fast segment evaluation.** Define the prefix sum of variances:

$$V_\ell(j) \triangleq \sum_{t=1}^{j} v_{\ell,t}, \qquad V_\ell(0) = 0. \tag{80}$$

Then for any segment $[i, j)$, we have

$$\sum_{t=i}^{j-1} v_{\ell,t} = V_\ell(j-1) - V_\ell(i-1). \tag{81}$$

Thus, $\text{SegCost}_\ell(i, j; b)$ can be computed in $\mathcal{O}(1)$ time.

**DP state.** Let $T \triangleq T_\ell$ for simplicity. We define the DP table $F[j, p]$ as the minimum achievable surrogate distortion for timesteps $1, \ldots, j$ using *exactly* budget $p$ (total assigned bits):

$$F[j, p] \triangleq \min_{\substack{\{b_t \in \mathcal{B}\}_{t=1}^{j} \\ \sum_{t=1}^{j} b_t = p}} \sum_{t=1}^{j} \kappa(b_t)\, v_{\ell,t}, \tag{82}$$

$$j \in \{0, \ldots, T\}, \quad p \in \{0, \ldots, B_\ell\}.$$

We set $F[0, 0] = 0$ and $F[0, p > 0] = +\infty$.

**Segment-based transition.** Consider the last segment ending at timestep $j$. Suppose this segment starts at timestep $i$ ($1 \le i \le j$) and uses bit-width $b \in \mathcal{B}$. Then the segment length is $(j - i + 1)$ and consumes budget $(j - i + 1) \cdot b$. The cost contributed by this segment is

$$\kappa(b) \sum_{t=i}^{j} v_{\ell,t}. \tag{83}$$

Hence, we obtain the recurrence:

$$F[j,p] = \min_{\substack{1 \le i \le j \\ b \in \mathcal{B} \\ p' = p-(j-i+1)b \ge 0}} \left( F[i-1, p'] + \kappa(b) \sum_{t=i}^{j} v_{\ell,t} \right). \tag{84}$$

Finally, since the constraint in (74) is $\le B_\ell$, the optimal value is

$$\min_{0 \le p \le B_\ell} F[T, p], \tag{85}$$

and the corresponding schedule is obtained by standard backtracking.

**Theorem C.3** (Optimality of the DP solution). *The DP recurrence* (84) *yields a globally optimal piecewise-constant schedule minimizing* (79) *under the budget constraint* (74).

*Proof.* We prove by induction on $j$ that $F[j,p]$ equals the optimal cost for timesteps $1, \ldots, j$ under exact budget $p$.

*Base case.* For $j = 0$, the empty schedule has cost 0 with budget 0, hence $F[0,0] = 0$ is optimal; any $p > 0$ is infeasible and $F[0,p] = +\infty$ is correct.

*Inductive step.* Assume the claim holds for all $(j', p')$ with $j' < j$. Consider an optimal segmented schedule for timesteps $1, \ldots, j$ with exact budget $p$. Let the last segment be $[i, j]$ with bit-width $b$. Then the prefix schedule on timesteps $1, \ldots, i-1$ must use budget $p' = p - (j-i+1)b$ and achieve its optimal cost $F[i-1, p']$ by the induction hypothesis. Adding the last segment cost gives a total cost equal to one candidate in (84), hence

$$F[j,p] \le F[i-1, p'] + \kappa(b) \sum_{t=i}^{j} v_{\ell,t}. \tag{86}$$

Conversely, any candidate in (84) corresponds to a valid segmented schedule formed by concatenating an optimal prefix schedule (by induction hypothesis) and a constant-bit last segment. Therefore, the minimum in (84) cannot be smaller than the true optimum. Thus, $F[j,p]$ equals the optimum for $(j, p)$. Applying the $\le B_\ell$ constraint via (85) completes the proof. $\square$

**Complexity.** The straightforward DP in (84) runs in $\mathcal{O}(T^2 B_\ell |\mathcal{B}|)$ time and $\mathcal{O}(T B_\ell)$ memory, which is efficient for typical diffusion timestep counts (Rombach et al., 2022). In practice, $T$ is at most a few dozen to a few hundred for diffusion scheduling, and $|\mathcal{B}|$ is small (e.g., $\{3, 4, 5\}$), making the per-layer optimization tractable. Further speedups are possible by restricting candidate boundaries or limiting the number of segments $M$.

### C.1.7. ALGORITHMIC IMPLEMENTATION

The following pseudocode summarizes the DP-based segmented temporal scheduling for a single layer $\ell$.

---

**Algorithm 2** VaTMP (Layer $\ell$): DP-Based Segmented Temporal Activation Bit Scheduling

---

**Require:** Timestep sensitivities $\{v_{\ell,t}\}_{t=1}^{T}$; candidate bit set $\mathcal{B}$; normalized coefficients $\{\kappa(b)\}_{b \in \mathcal{B}}$; budget $B_\ell = \lfloor B_{\text{target}}^{\text{act}} T \rfloor$.

**Ensure:** A piecewise-constant schedule $\{b_{\ell,t}\}_{t=1}^{T}$ with $\sum_{t=1}^{T} b_{\ell,t} \le B_\ell$.

1: Compute prefix sums $V(0) \leftarrow 0$, $V(j) \leftarrow \sum_{t=1}^{j} v_{\ell,t}$ for $j = 1, \ldots, T$.
2: Initialize DP table $F[0,0] \leftarrow 0$ and $F[0, p > 0] \leftarrow +\infty$.
3: Initialize parent pointers $\Pi[j,p] \leftarrow \varnothing$.
4: **for** $j \leftarrow 1$ **to** $T$ **do**
5:    **for** $p \leftarrow 0$ **to** $B_\ell$ **do**
6:       $F[j,p] \leftarrow +\infty$.
7:       **for** $i \leftarrow 1$ **to** $j$ **do**
8:          $S \leftarrow V(j) - V(i-1)$ $\{S = \sum_{t=i}^{j} v_{\ell,t}\}$
9:          **for all** $b \in \mathcal{B}$ **do**
10:             $p' \leftarrow p - (j-i+1) \cdot b$.
11:             **if** $p' \ge 0$ **and** $F[i-1, p'] < +\infty$ **then**
12:                cand $\leftarrow F[i-1, p'] + \kappa(b) \cdot S$.
13:                **if** cand $< F[j,p]$ **then**
14:                   $F[j,p] \leftarrow$ cand.
15:                   $\Pi[j,p] \leftarrow (i, b, p')$.
16:                **end if**
17:             **end if**
18:          **end for**
19:       **end for**
20:    **end for**
21: **end for**
22: Select $p^\star \leftarrow \arg\min_{0 \le p \le B_\ell} F[T, p]$.
23: Backtrack from $(T, p^\star)$ using $\Pi$ to recover segments and fill $\{b_{\ell,t}\}_{t=1}^{T}$.
24: **return** $\{b_{\ell,t}\}_{t=1}^{T}$.

---

### C.1.8. DISCUSSION: WHY INTRA-LAYER TEMPORAL MP IS STABLE

Finally, we highlight why VaTMP performs allocation *within each layer* rather than across layers. Different layer families in diffusion SR backbones (e.g., attention (Vaswani et al., 2017), MLP, modulation, and control branches) exhibit heterogeneous activation statistics and functional roles, making cross-layer variance comparisons less reliable. This design contrasts with prior cross-layer mixed-precision methods (Dong et al., 2019; Zhao et al., 2024). In contrast, within a fixed layer $\ell$, the function class and normalization behavior remain consistent, and the temporal evolution of $v_{\ell,t}$ provides a stable, monotonic sensitivity cue. Together with the variance-scaling law in Lemma C.1, this yields a principled and robust scheduling criterion that aligns high precision with high-variance timesteps while satisfying the same average-bit budget.

### C.2. Temporal Mixed-Precision Scheduling for Activations

This section presents a detailed analysis of the activation bit-width scheduling produced by our VaTMP strategy

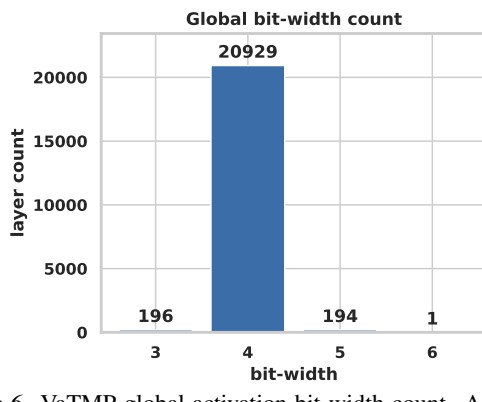

Figure 6. VaTMP global activation bit-width count. Across all layer–timestep pairs, 4-bit dominates the allocation, while only a small fraction is adjusted to 3/5/6 bits for budget redistribution.

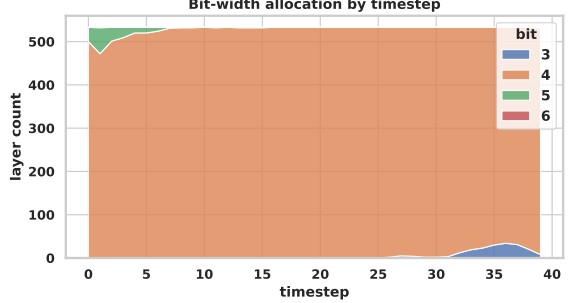

Figure 7. VaTMP bit allocation by diffusion timestep. While 4-bit remains the default precision throughout the trajectory, VaTMP allocates slightly higher precision at early timesteps and shifts a small portion to lower precision at late timesteps, reflecting timestep-dependent activation sensitivity.

across diffusion timesteps. VaTMP performs *intra-layer* mixed-precision allocation over time, selecting activation bit-widths from a discrete candidate set $\{2, 3, 4, 5, 6, 7, 8\}$ under a global activation budget, so as to minimize the overall quantization distortion along the denoising trajectory (Li et al., 2023; So et al., 2023).

**Global allocation is strongly centered around 4-bit.** We first report the global usage frequency of each activation precision in Fig. 6. Across all layer–timestep pairs, VaTMP assigns 4-bit to 20,929 cases, while only a small fraction deviates to 3-bit (196 cases), 5-bit (194 cases), or 6-bit (1 case). This highly concentrated distribution indicates that VaTMP preserves a stable 4-bit backbone for most activations, and only makes sparse temporal adjustments where the predicted distortion benefit justifies spending (or saving) extra bits, consistent with timestep-dependent sensitivity observed in diffusion quantization (He et al., 2023; Shang et al., 2023).

**Timestep-wise redistribution: higher bits early, lower bits late.** Fig. 7 shows the stacked bit-width counts over timesteps. While 4-bit remains dominant throughout the entire trajectory, VaTMP exhibits a clear tendency to allocate more 5/6-bit activations at early timesteps and gradually

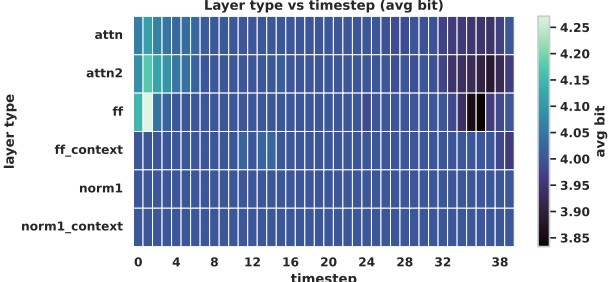

Figure 8. VaTMP scheduling by layer type. Heatmap of the average activation bit-width grouped by module type across diffusion timesteps. MLP/attention-related modules show more temporal variation, whereas normalization-related layers remain nearly constant, indicating weaker timestep sensitivity.

shift a small portion to 3-bit at later timesteps. This pattern suggests that the early denoising stage is more sensitive to activation quantization noise (typically associated with larger activation variance), and thus benefits from higher precision, whereas late-stage refinement can tolerate more aggressive compression without harming perceptual fidelity.

**Temporal sensitivity depends on layer type.** Fig. 8 groups scheduled precision by layer type. We observe that ff and attn2 show the strongest temporal variation: they receive higher average bits early and decrease steadily toward later timesteps. In contrast, norm1 and norm1_context stay nearly constant around 4 bits, indicating weak temporal sensitivity. This matches the role of each module: MLP and attention projections dominate feature transformation and token mixing, making them more sensitive to timestep-dependent noise, while normalization layers are more robust, requiring less temporal reallocation.

**VaTMP concentrates variability on a small subset of layers.** Finally, Fig. 9 visualizes the top-variable layers across timesteps. A key observation is that most high-variability layers correspond to attention control/value projections (e.g., to_v_control, to_k_control) and a small number of output/MLP projections. These layers receive noticeably higher precision at early timesteps (often 5–6 bits) and are aggressively reduced to 3-bit at the end of the trajectory. This indicates that VaTMP does *not* uniformly change precision across the network; instead, it identifies a narrow set of temporally sensitive layers and reallocates bits in a highly selective manner, providing an interpretable and efficient temporal mixed-precision schedule.

**Case study: per-block and per-layer temporal schedules.** We further present a case study by visualizing the detailed bit-width scheduling for several representative blocks and layers. As shown in Fig. 10, VaTMP keeps the majority of layers within each block at a stable 4-bit across timesteps, while concentrating most temporal variation on a small subset of *control* projections and a few high-impact transfor-

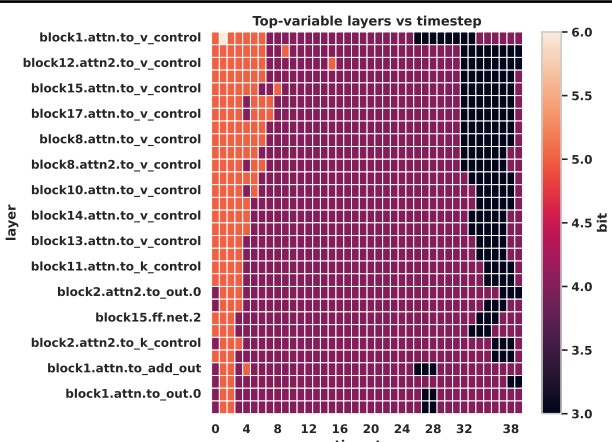

*Figure 9.* Top-variable layers in VaTMP across timesteps. VaTMP concentrates most temporal precision variation on a small subset of layers (often attention control/value projections), assigning higher precision early and reducing precision later, which yields an interpretable and efficient temporal mixed-precision schedule.

mations. In particular, `to_v_control` (and occasionally `to_k_control`/`to_q_control`) consistently receives elevated precision at very early timesteps (typically 5 bits, and up to 6 bits in block 1), indicating strong sensitivity of these control-conditioned value pathways when the denoising process starts from high-noise states. As the trajectory proceeds, these layers quickly return to the 4-bit backbone and remain stable through the mid-stage, suggesting that a moderate precision is sufficient once the signal becomes more structured. Toward late timesteps, VaTMP aggressively reduces these same layers to 3 bits in multiple blocks, reflecting lower temporal sensitivity during the final refinement stage and enabling budget redistribution without harming perceptual quality. Notably, this late-stage reduction is selective and does not affect most attention/MLP projections, highlighting that VaTMP avoids uniform compression and instead targets temporally redundant components.

To illustrate temporal behavior per layer, Fig. 11 plots the bit-width timeline for representative `to_v_control` layers. Across blocks, we observe a consistent "high–stable–low" schedule: an early high-precision phase (e.g., 5! →!6 in block 1), a long plateau at 4 bits, and a late reduction to 3 bits. Notably, some layers show brief mid-trajectory spikes (e.g., a quick return to 5 bits), indicating that VaTMP can capture localized sensitivity peaks without enforcing monotonic schedules. Moreover, for certain layers VaTMP raises precision back to 4 bits at the final step—a lightweight safeguard to preserve last-stage reconstruction quality within the same global activation budget. Overall, these visualizations confirm VaTMP yields a sparse yet structured temporal allocation: most layer–timestep pairs remain at 4 bits, while only a few temporally sensitive control pathways are dynamically adjusted to best trade distortion for efficiency.

## D. Overall PTQ Pipeline of Q-DiT4SR

We summarize the full post-training quantization pipeline of **Q-DiT4SR**, which combines hierarchical SVD-based weight reconstruction (H-SVD), variance-aware cross-layer weight mixed precision (VaSMP), and variance-aware intra-layer temporal activation mixed precision (VaTMP). The resulting quantized model preserves the dominant FP information flow with lightweight full-precision branches while allocating limited bit budgets in a distortion-aware manner, enabling efficient deployment of DiT-based SR models (Duan et al., 2025; Ai et al., 2024).

## E. Additional Visual Comparison

Additional visual comparisons are provided in Fig. 12 to Fig. 17. These figures show results of our method Q-DiT4SR on the DrealSR (Wei et al., 2020), RealSR (Cai et al., 2019), and RealLQ250 (Ai et al., 2024) datasets under both W4A6 and W4A4 settings, compared with SVDQuant (Li et al., 2024), QuaRot (Ashkboos et al., 2024), Q-DiT (Chen et al., 2025), PTQ4DiT (Wu et al., 2024), PassionSR (Zhu et al., 2025), FlatQuant (Sun et al., 2025) and QueST (Wang et al., 2025).

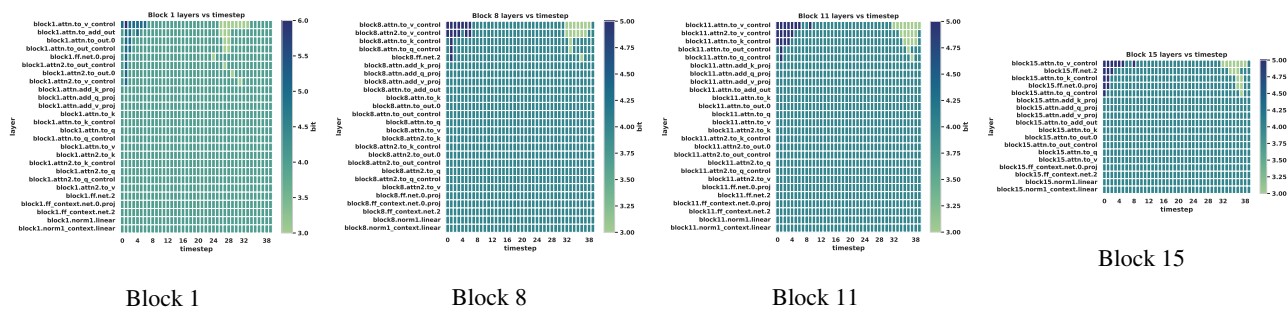

Block 1          Block 8          Block 11

*Figure 10.* Selected blocks: per-layer activation bit-width scheduling across diffusion timesteps. VaTMP keeps most layers at a stable 4-bit backbone, while concentrating temporal variation on a few control-related projections (e.g., `to_v_control`) that receive higher precision at early timesteps and are reduced at late timesteps for budget redistribution.

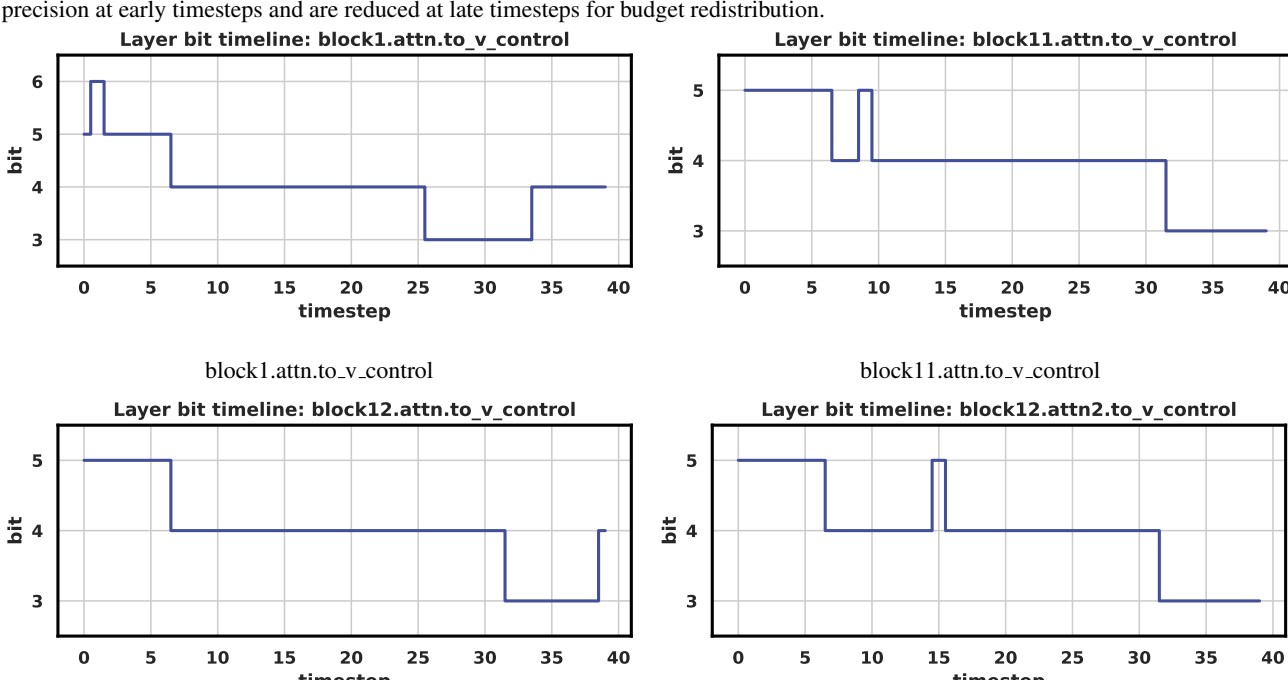

block1.attn.to_v_control          block11.attn.to_v_control

block12.attn.to_v_control          block12.attn2.to_v_control

*Figure 11.* Layer-wise bit-width timelines for representative `to_v_control` projections. VaTMP exhibits a consistent "high–stable–low" temporal schedule, with early high precision, a long 4-bit plateau, and late-stage reduction to 3 bits, occasionally with localized spikes capturing mid-trajectory sensitivity peaks.

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

---

### Algorithm 3 Overall PTQ Pipeline of Q-DiT4SR

---

**Require:** Pretrained DiT-based SR model with linear layers $\{\mathbf{W}^{(\ell)}\}$; normalized Hadamard matrix $\mathbf{H}_n$; target SVD rank $r{=}32$; feasible block sizes $\mathcal{S}$; weight-bit candidates $\mathcal{B}_w{=}\{2,3,4,5,6,7,8\}$; activation-bit candidates $\mathcal{B}_a{=}\{2,3,4,5,6,7,8\}$; target average weight bits $B_{\text{target}}$; target average activation bits $B_{\text{target}}^{\text{act}}$; uniform quantizer $Q_{\text{uni}}(\cdot)$ pre-optimized for $\mathcal{N}(0,1)$; small LR calibration set $\mathcal{D}_{\text{LR}}$ for activation statistics.

**Ensure:** Quantized model with H-SVD weight reconstruction, VaSMP weight bit-widths $\{b_\ell\}$, and VaTMP activation schedules $\{b_{\ell,t}\}$.

1: **(I) Weight-side reconstruction and mixed precision (H-SVD + VaSMP).**
2: **for** each active linear layer $\ell \in \mathcal{A}$ with weight $\mathbf{W}^{(\ell)} \in \mathbb{R}^{out \times in}$ **do**
3:    **Hadamard transform:** $\mathbf{W}_{\text{H}}^{(\ell)} \leftarrow \mathbf{W}^{(\ell)} \mathbf{H}_n$.
4:    **Global SVD branch:** $\mathbf{W}_{\text{SVD-G}}^{(\ell)} \leftarrow \text{TruncatedSVD}(\mathbf{W}_{\text{H}}^{(\ell)}, r)$.
5:    **Residual:** $\mathbf{W}_{\text{res}}^{(\ell)} \leftarrow \mathbf{W}_{\text{H}}^{(\ell)} - \mathbf{W}_{\text{SVD-G}}^{(\ell)}$.
6:    **Budget-matched local branch:**
7:      choose $(s_o, s_i) \in \mathcal{S}$ satisfying the budget-matching constraint.
8:    **Localized SVD (blockwise rank-1):**
9:      $\mathbf{W}_{\text{SVD-L}}^{(\ell)} \leftarrow \text{Assemble}\big(\{\text{Rank1SVD}(\mathbf{W}_{\text{res}}^{(\ell)}[p,q])\}_{p,q}\big)$.
10:    **Offline variance stats:** compute $N_\ell$ and $\bar{\sigma}_\ell^2$ in the Hadamard domain.
11: **end for**
12: **VaSMP bit allocation:** compute continuous $b_\ell^*$ and discretize via greedy gains to obtain integer $b_\ell \in \mathcal{B}_w$.
13: **Quantize weight residuals:**
14: **for** each active layer $\ell \in \mathcal{A}$ **do**
15:    $\tilde{\mathbf{W}}_{\text{res}}^{(\ell)} \leftarrow \mathbf{W}_{\text{res}}^{(\ell)} - \mathbf{W}_{\text{SVD-L}}^{(\ell)}$.
16:    **Per-output-channel quantization:** $Q_w(\tilde{\mathbf{W}}_{\text{res}}^{(\ell)}) \leftarrow \sigma_o \cdot Q_{\text{uni}}(\tilde{\mathbf{W}}_{\text{res}}^{(\ell)}/\sigma_o)$.
17:    **Reconstruct quantized weight:** $\hat{\mathbf{W}}^{(\ell)} \leftarrow \big(\mathbf{W}_{\text{SVD-G}}^{(\ell)} + \mathbf{W}_{\text{SVD-L}}^{(\ell)} + Q_w(\tilde{\mathbf{W}}_{\text{res}}^{(\ell)})\big)\mathbf{H}_n^\top$.
18: **end for**

19: **(II) Activation-side temporal mixed precision (VaTMP).**
20: **Collect temporal sensitivity:**
21: **for** each layer $\ell \in \mathcal{A}$ and timestep $t$ **do**
22:    Run a forward pass on $\mathcal{D}_{\text{LR}}$ with fixed quantized weights $\{\hat{\mathbf{W}}^{(\ell)}\}$.
23:    Compute timestep variance proxy $v_{\ell,t} \leftarrow \text{mean\_token\_var}(\ell, t)$.
24: **end for**
25: **Precompute normalized distortion coefficients:** $\kappa(b)$ for all $b \in \mathcal{B}_a$.
26: **for** each layer $\ell \in \mathcal{A}$ **do**
27:    Define budget $B_\ell \leftarrow \lfloor B_{\text{target}}^{\text{act}} T_\ell \rfloor$.
28:    Solve the segmented DP schedule: $\{b_{\ell,t}\}_{t=1}^{T_\ell} \leftarrow \arg\min \sum_{t=1}^{T_\ell} \kappa(b_{\ell,t})v_{\ell,t}$ s.t. $\sum_{t=1}^{T_\ell} b_{\ell,t} \leq B_\ell$ using segment costs and piecewise-constant schedules.
29: **end for**

30: **(III) Deployment (quantized inference).**
31: **for** each diffusion timestep $t$ **do**
32:    **for** each layer $\ell$ **do**
33:      **Activation quantization (Hadamard Gaussianization):** $\hat{\mathbf{x}} \leftarrow Q_G(\mathbf{x}; b_{\ell,t})$.
34:      **Linear forward with H-SVD weight reconstruction:** $\mathbf{y} \leftarrow Q_w(\tilde{\mathbf{W}}_{\text{res}}^{(\ell)})\hat{\mathbf{x}} + (\mathbf{W}_{\text{SVD-G}}^{(\ell)} + \mathbf{W}_{\text{SVD-L}}^{(\ell)})\mathbf{H}_n^\top \mathbf{x}$.
35:    **end for**
36: **end for**
37: **return** Quantized model with $\{\hat{\mathbf{W}}^{(\ell)}\}$, $\{b_\ell\}$, and $\{b_{\ell,t}\}$.

---

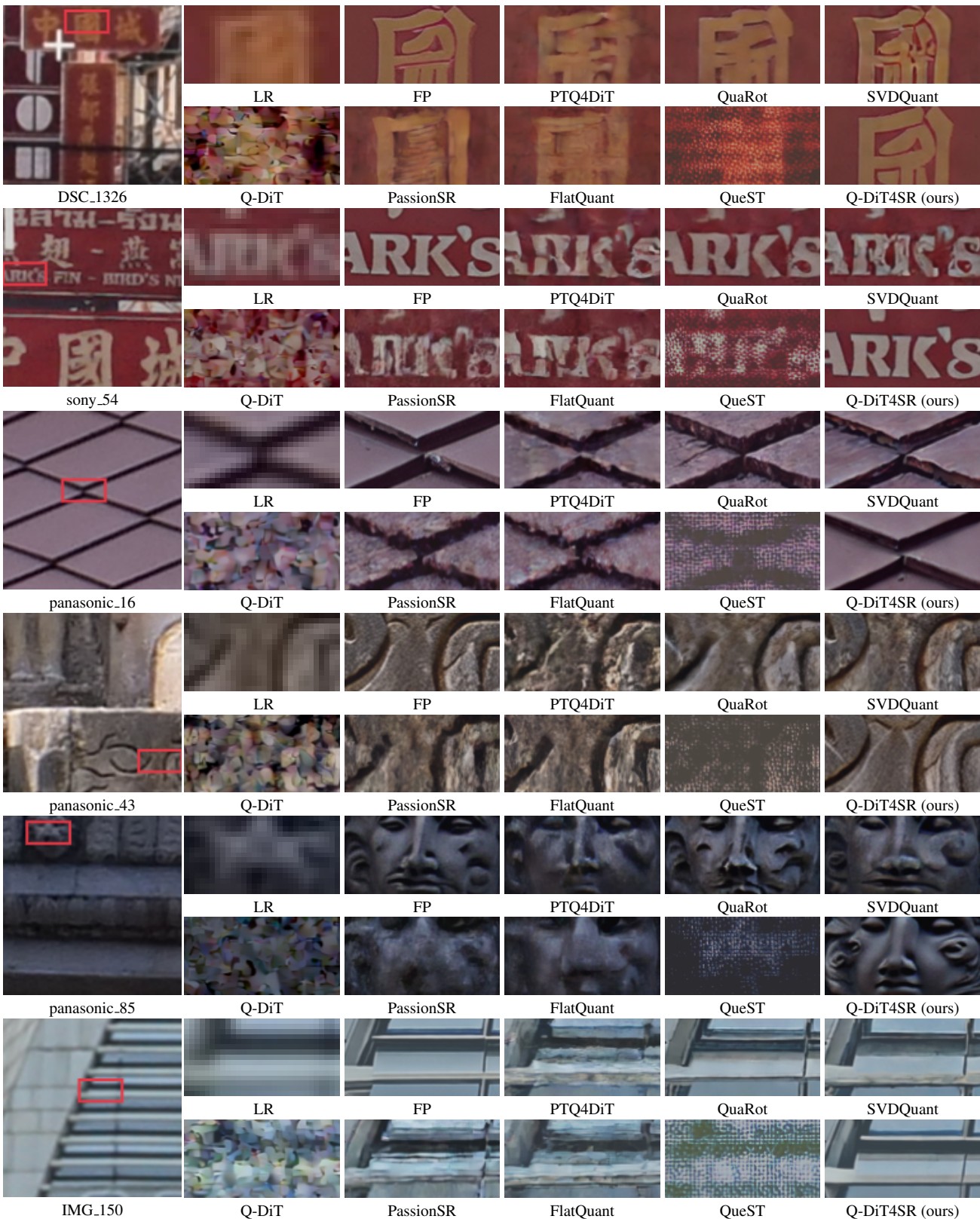

*Figure 12.* Visual comparison (×4) of SR results on the **DrealSR** dataset under the **W4A6** setting, comparing our method Q-DiT4SR with other competitive quantization baselines and the FP model. Q-DiT4SR gains significant visual advantages over other methods.

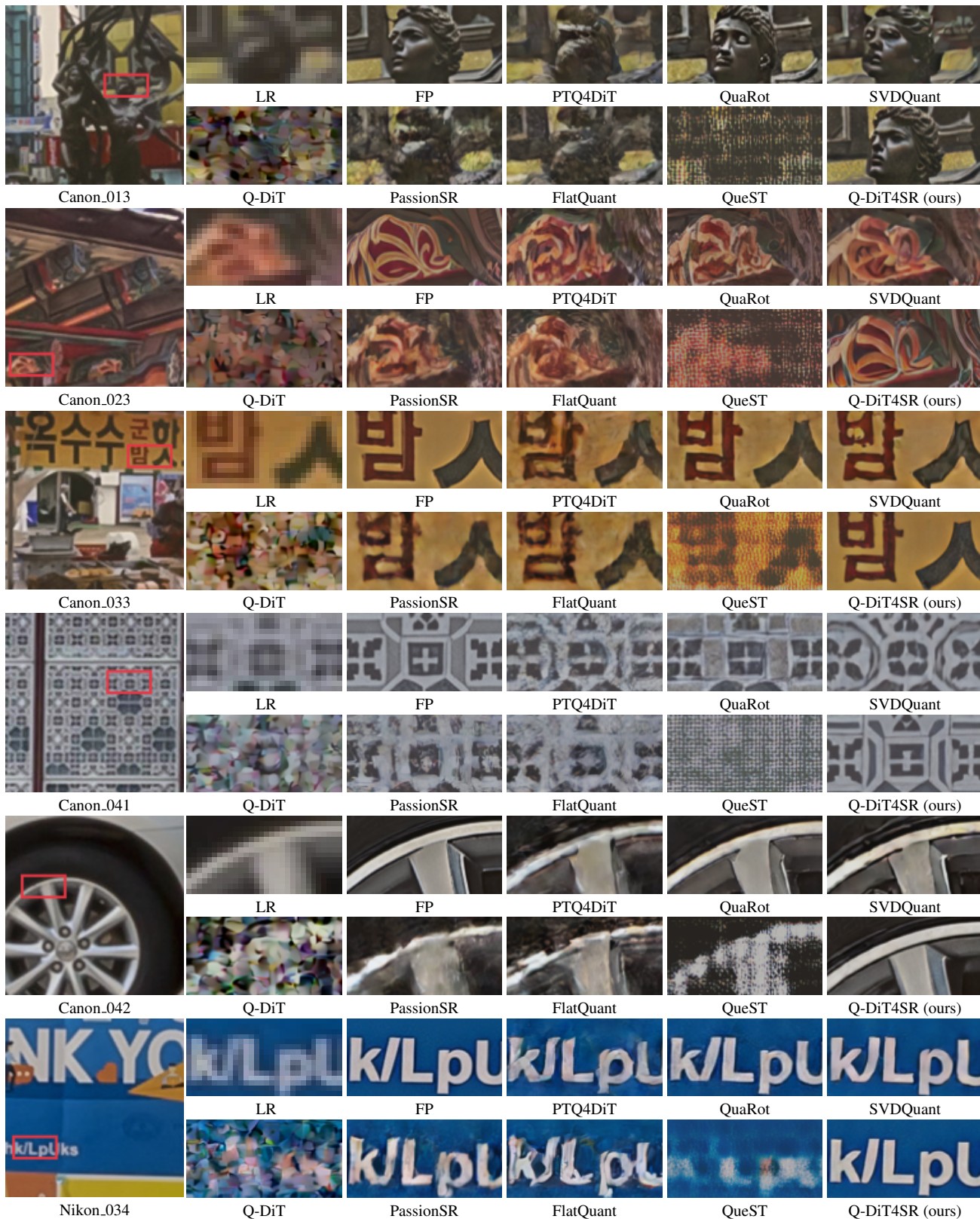

*Figure 13.* Visual comparison (×4) of SR results on the **RealSR** dataset under the **W4A6** setting, comparing our method Q-DiT4SR with other competitive quantization baselines and the FP model. Q-DiT4SR gains significant visual advantages over other methods.

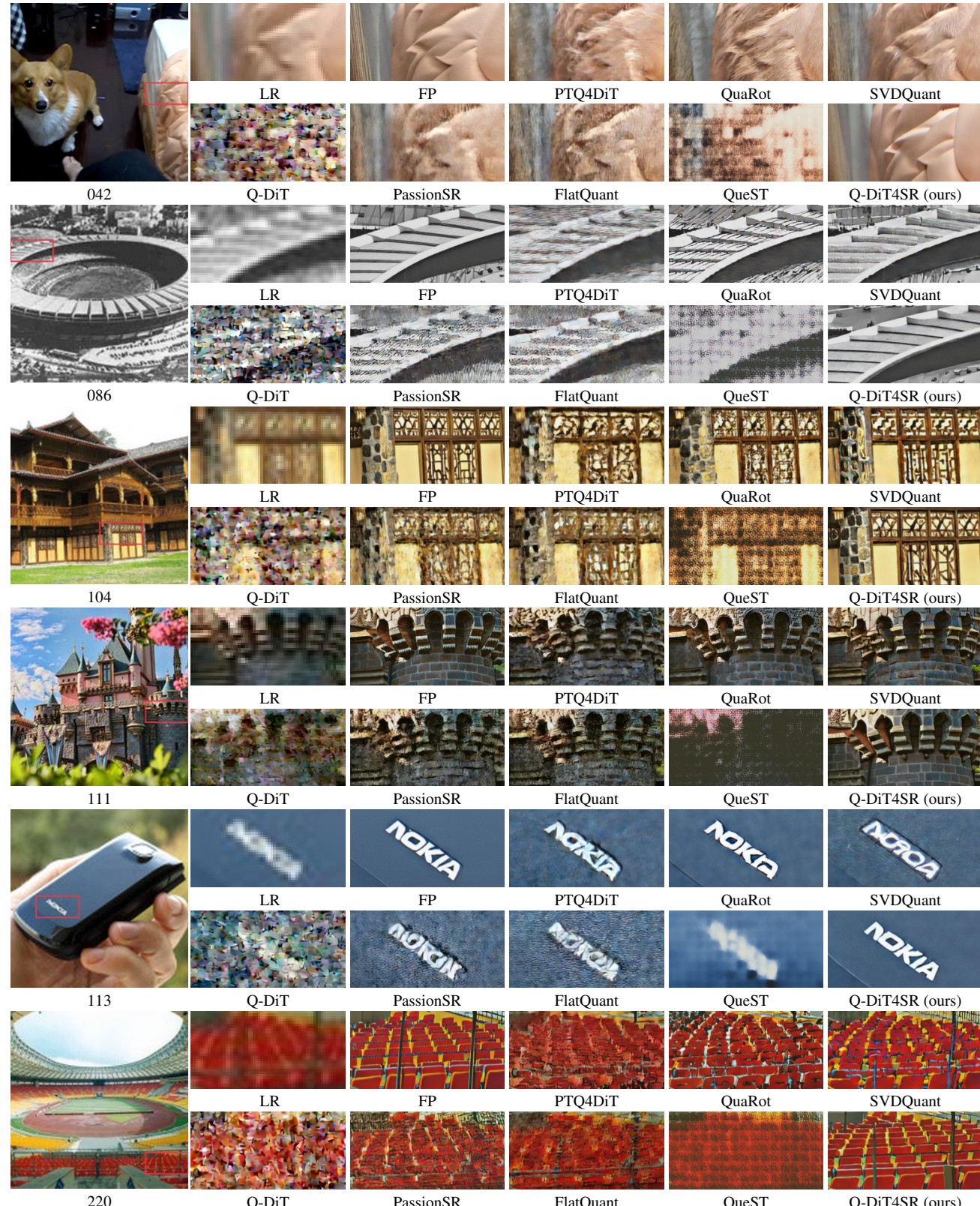

*Figure 14.* Visual comparison (×4) of SR results on the **RealLQ250** dataset under the **W4A6** setting, comparing our method Q-DiT4SR with other competitive quantization baselines and the FP model. Q-DiT4SR gains significant visual advantages over other methods.

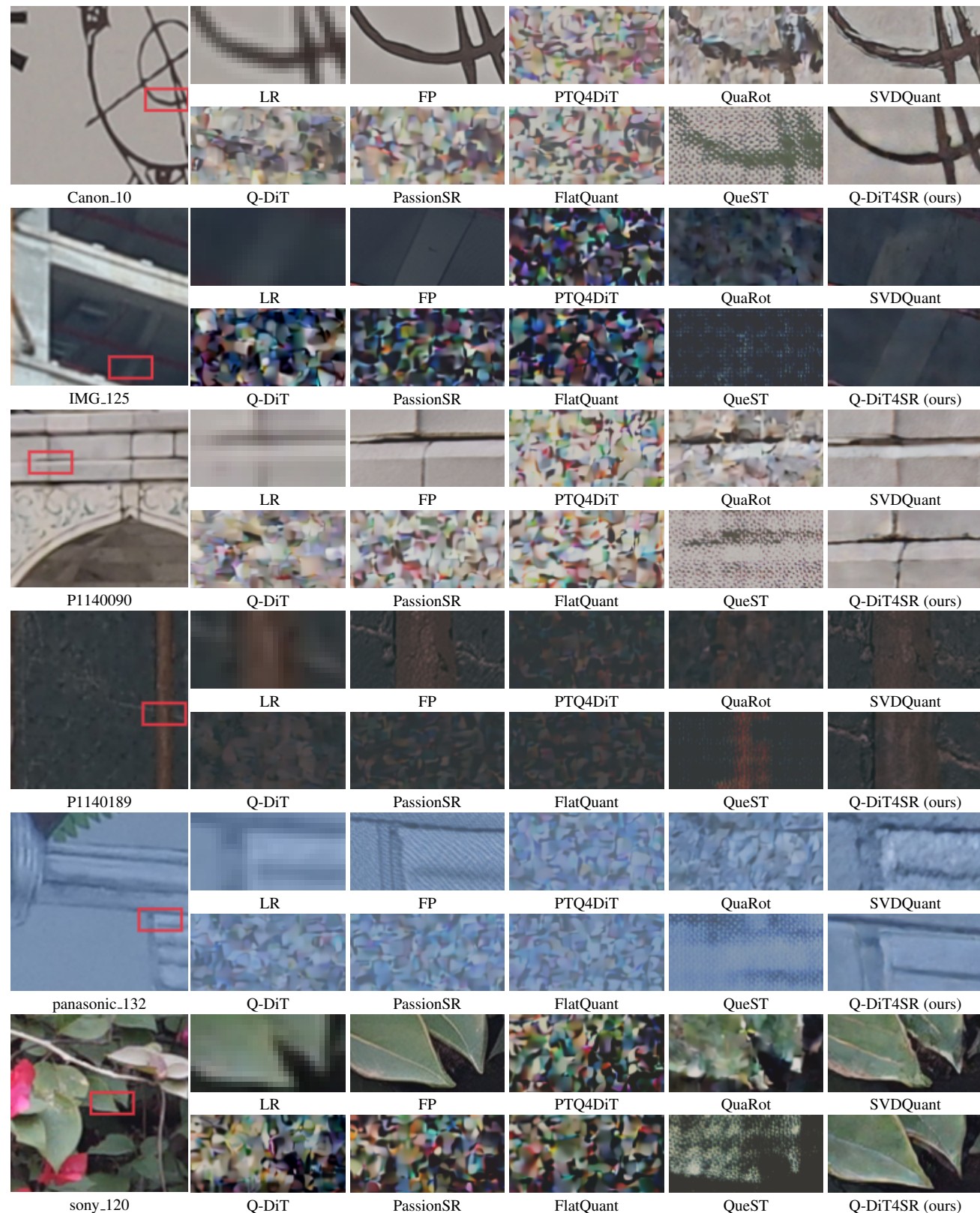

*Figure 15.* Visual comparison (×4) of SR results on the **DrealSR** dataset under the **W4A4** setting, comparing our method Q-DiT4SR with other competitive quantization baselines and the FP model. Q-DiT4SR gains significant visual advantages over other methods.

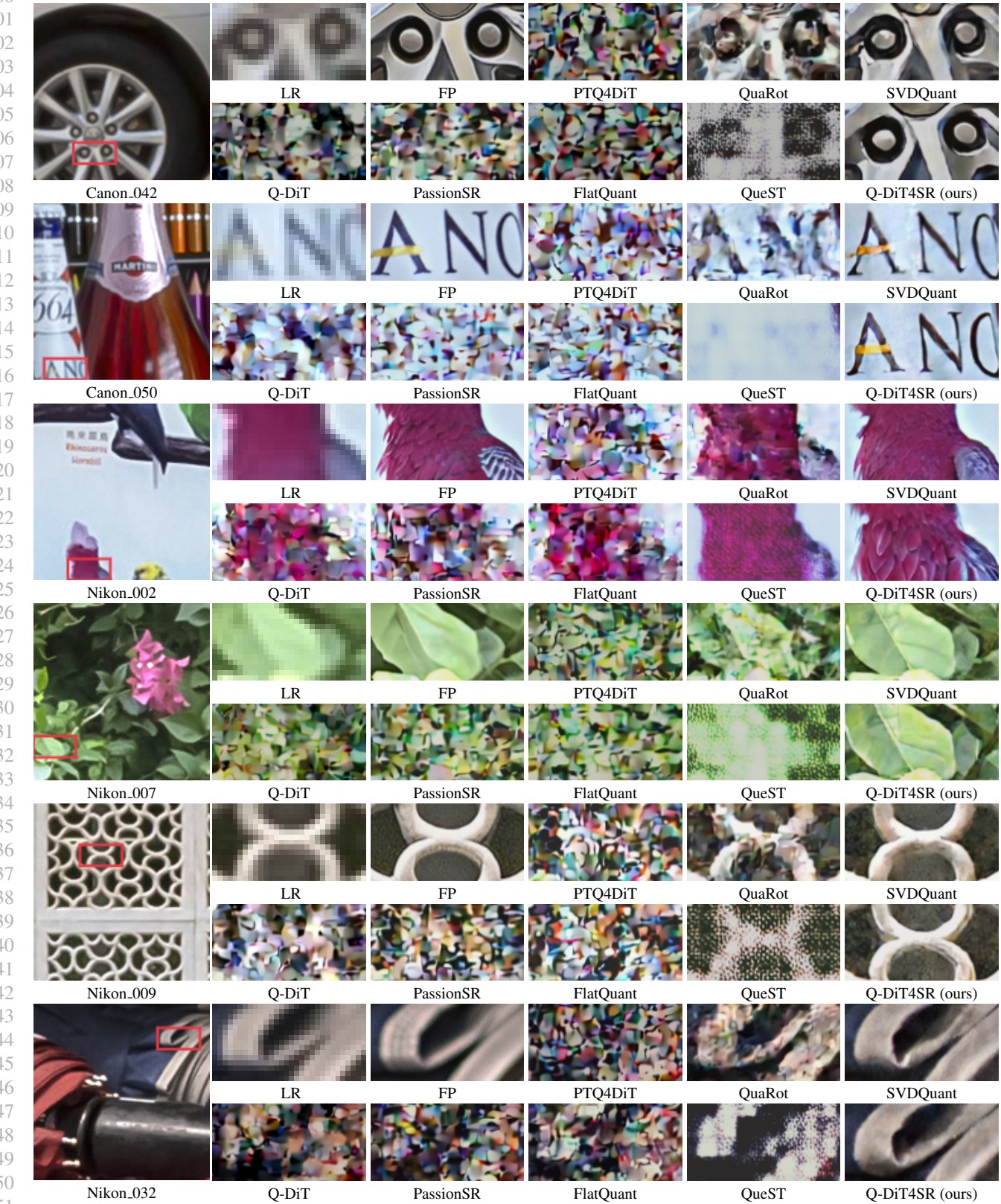

*Figure 16.* Visual comparison (×4) of SR results on the **RealSR** dataset under the **W4A4** setting, comparing our method Q-DiT4SR with other competitive quantization baselines and the FP model. Q-DiT4SR gains significant visual advantages over other methods.

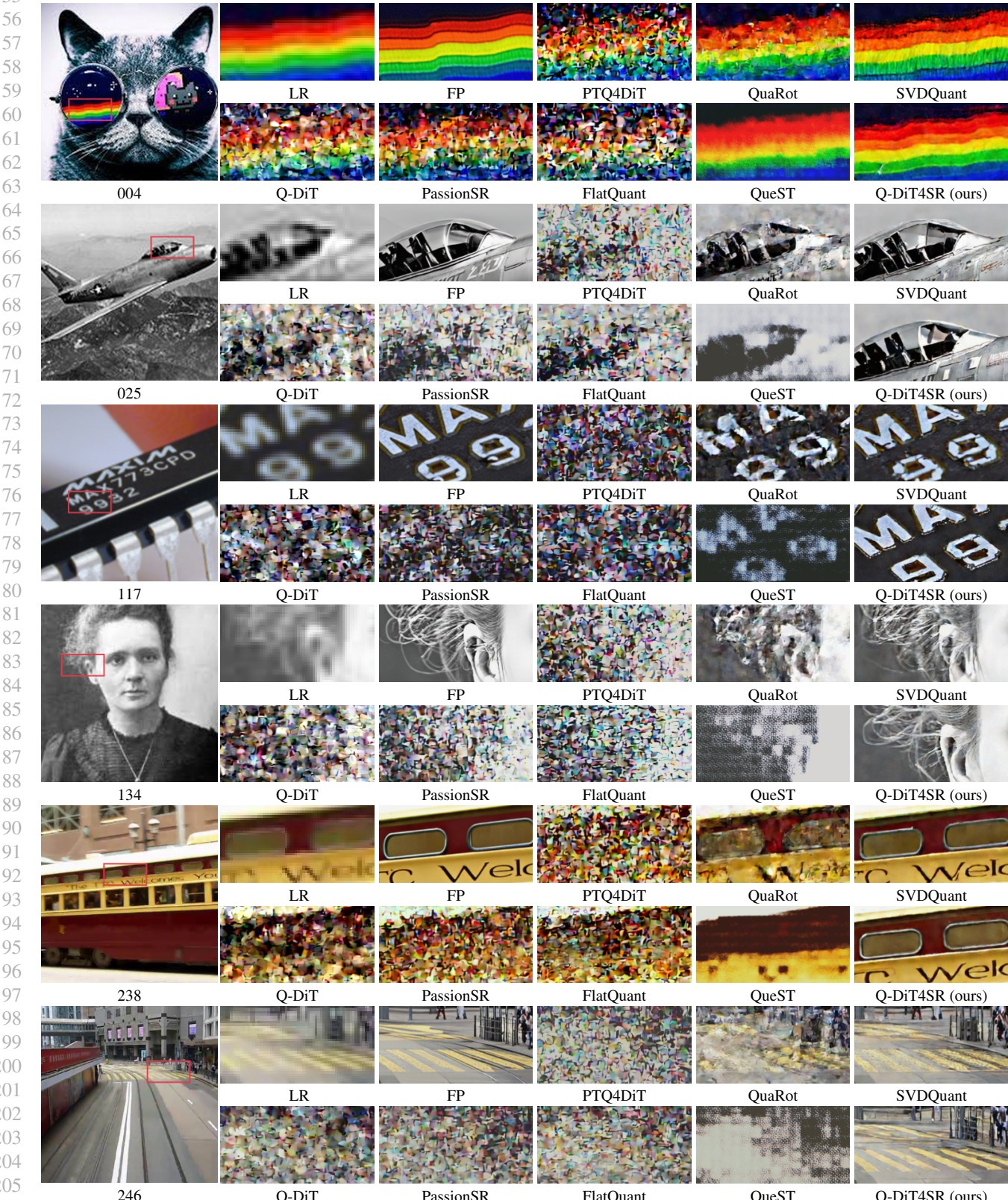

*Figure 17.* Visual comparison (×4) of SR results on the **RealLQ250** dataset under the **W4A4** setting, comparing our method Q-DiT4SR with other competitive quantization baselines and the FP model. Q-DiT4SR gains significant visual advantages over other methods.