# OpenReview forum: "Q-DiT4SR: Exploration of Detail-Preserving Diffusion Transformer  Quantization for Real-World Image Super-Resolution"
_ICML.cc/2026/Conference — ICML 2026 regular_

### Official Review · Reviewer_jgvJ · 2026-02-17

**Soundness:** 3
**Presentation:** 3
**Significance:** 4
**Originality:** 3
**Overall Recommendation:** 5
**Confidence:** 3

**Summary:**

The authors propose Q-DiT4SR, a framework comprising three key components: 1) H-SVD, which decomposes weights into a global low-rank branch and a local block-wise rank-1 branch to preserve fine-grained textures; 2) VaSMP, a data-free strategy that allocates weight bit-widths across layers based on variance; and 3) VaTMP, a strategy that schedules activation bit-widths across diffusion timesteps with minimal calibration. The method achieves SOTA performance under aggressive W4A6 and W4A4 settings.

**Compliance With Llm Reviewing Policy:**

Affirmed.

**Key Questions For Authors:**

1. Can the authors provide actual inference latency on a standard GPU for generating one image? Theoretical BOPs reduction often does not translate linearly to speedup, especially with mixed-precision kernels. need comparison with other methods.
2. How do the authors envision the deployment of arbitrary bit-widths (e.g., 3-bit, 5-bit) generated by VaSMP? Are these relying on a specific compiler or hardware support, or is this primarily a simulation of potential savings?
3. Does the H-SVD and variance-based allocation generalize to other low-level vision tasks (e.g., deblurring) using DiTs, or is it strictly beneficial only for the high-frequency nature of SR?

**Limitations:**

The primary limitation may lies in the hardware realizability of the proposed mixed-precision scheme. While reducing theoretical bit-ops is beneficial, current general-purpose GPUs are optimized for specific precisions. Supporting fine-grained mixed precision (switching between 3-bit and 5-bit per layer or timestep) requires specialized kernel engineering or hardware to realize the reported speedups. Without this, the method serves more as a theoretical exploration of redundancy in DiT-SR models rather than an immediately deployable acceleration solution.

**Strengths And Weaknesses:**

Strengths:
1. The first PTQ framework specifically optimized for DiT-based SR.
2. The proposed Hierarchical SVD is a clever decomposition. By combining a global approximation with local block-wise residuals, it effectively addresses the "detail loss" problem in standard quantization
3. The proposed mixed-precision strategies are well-motivated by the analysis of weight and activation variance distributions.
4. The quantitative results show significant improvements over existing baselines in perceptual metrics, especially in the challenging W4A4 setting.

Weaknesses:

1. The paper claims a "60× reduction in operations," but this is a theoretical metric. Implementing mixed-precision (e.g., varying bit-widths like 3-bit, 5-bit across layers or timesteps) on standard GPUs is difficult to accelerate without specialized kernels.

---

> ### Author Rebuttal · Authors · 2026-03-30
>
> `Q3-1` Can the authors provide actual inference latency on a standard GPU for generating one image? Theoretical BOPs reduction often does not translate linearly to speedup, especially with mixed-precision kernels. need comparison with other methods.
>
> `A3-1` We first clarify a numerical error in the manuscript. The Ops value of the FP model was mistakenly reported as **170,476**, while the correct value is **17,047.6** (rounded to **17,048**). The corresponding theoretical speedup should therefore be corrected to **17,048 / 2,776 ≈ 6.14×**. If accepted, we will provide the detailed calculation procedure in the camera-ready version.
>
> From an implementation perspective, our deployment is efficient (tested on RTX-4090 48GB, batch size = 16, one-step forward). The H-SVD global branch uses branch-parallel execution (similar to Nunchaku), while the local branch is implemented with an efficient Triton kernel inspired by Monarch matrices, resulting in minimal runtime overhead.
>
> We agree that theoretical BOP reduction alone is insufficient. Our measurements show meaningful practical speedup: compared with **W4A4 SVDQuant** (~**4.8×**), our method achieves ~**4.5×** end-to-end speedup with better visual quality, indicating it is not merely theoretical. We also observe substantial memory reduction (peak GPU memory: **FP 15085.99 MiB**, **SVDQuant 3722.83 MiB**, **Ours 3974.64 MiB**). The gap to theoretical speedup arises because only the **quantized linear layers** achieve high acceleration (**8.99×**, 1580.91 → 175.88 ms), while **nonlinear / unquantized components** improve much less (**2.97×**, 1518.39 → 510.62 ms), making them the dominant bottleneck.
>
> Due to rebuttal time constraints, we have not implemented native W3A3 / W5A5 kernels. However, recent work supports arbitrary-/irregular-precision inference, suggesting the gap mainly stems from kernel maturity, not the quantization principle. We will explore adapting our method to such backends in future work.
>
>
>
>
>
> `Q3-2` Deployment and practicality: How can the arbitrary bit-widths (e.g., 3-bit, 5-bit) produced by VaSMP be deployed in practice? Does the method rely on specific hardware or kernels, and to what extent are the reported gains realizable on current GPUs?
>
> `A3-2` We thank the reviewer for this important point. We agree that the reported operation reduction should be interpreted as a theoretical compute proxy, not a direct wall-clock speedup. Our contribution is to show that the proposed PTQ design improves the accuracy–efficiency trade-off for DiT-based Real-ISR by reducing low-bit computation while preserving quality. Our method is hardware-friendly, assigning bit-widths to standard linear layers without introducing special operators, and can in principle be supported by emerging arbitrary-/irregular-precision kernels (e.g., Quant-LLM, ABQ-LLM). Due to time constraints, we have not yet explored this direction in depth. We thus view VaSMP/VaTMP as producing general precision profiles rather than purely theoretical savings, and will further explore practical and open-source kernel integration in future work.
>
>
>
>
>
> `Q3-3` Does the H-SVD and variance-based allocation generalize to other low-level vision tasks (e.g., deblurring) using DiTs, or is it strictly beneficial only for the high-frequency nature of SR?
>
> `A3-3` We thank the reviewer for this important question. As discussed in our response to **Reviewer NhyH（`A2-1`）** , we additionally evaluated our method on **DreamClear**. The table below shows the results of our quantization method on DreamClear across other datasets.
>
> |    Dataset    |  Metric   | DreamClear (FP) | Q-DiT4SR (W4A6) | Q-DiT4SR (W4A4) |
> | :-----------: | :-------: | :--------: | :-------------: | :-------------: |
> |  **DrealSR**  |  LPIPS ↓  |   0.3601   |     0.4583      |     0.4299      |
> |               |  MUSIQ ↑  |   43.38    |      49.13      |      43.51      |
> |               | MANIQA ↑  |   0.3173   |     0.3897      |     0.3241      |
> |               | ClipIQA ↑ |   0.3761   |     0.4330      |     0.3650      |
> |               |  LIQE ↑   |   2.308    |      2.842      |      2.209      |
> | **RealLQ250** |  MUSIQ ↑  |   66.46    |      65.44      |      61.16      |
> |               | MANIQA ↑  |   0.4401   |     0.3777      |     0.3426      |
> |               | ClipIQA ↑ |   0.5004   |     0.4596      |     0.4046      |
> |               |  LIQE ↑   |   3.708    |      3.350      |      2.930      |
>
> Beyond this empirical result, our method is **not SR-specific by design**: H-SVD and VaSMP/VaTMP explicitly target **quantization error minimization** and are thus expected to generalize to other DiT-based low-level vision tasks. Due to time/resource constraints and limited open-source DiT models for such tasks, we leave this as future work. We would also be happy to conduct additional experiments if the reviewer could suggest specific methods or publicly available repositories for other DiT-based low-level tasks.

---

> > ### Author Rebuttal · Reviewer_jgvJ · 2026-04-02
> >
> > For Q3-1 and Q3-2, my concerns persist.
> >
> > The efficiency claims remain theoretical because:
> > 1. The lack of end-to-end, wall-clock latency benchmarks on standard GPU configurations.
> > 2. The method is not yet "hard ware-friendly" in practice, as it relies on future, non-existent kernel integration for arbitrary-precision support.
> >
> > Since the practical deployability is currently unverified and hinges on future work, I will maintain my score.

---

> > > ### Author Response · Authors · 2026-04-03
> > >
> > > We sincerely thank the reviewer for the follow-up and the opportunity to clarify the practical deployment of our framework.
> > >
> > > To directly address the concerns regarding theoretical efficiency and wall-clock latency, we have implemented a hardware-ready mixed-precision version on a standard RTX-4090 48GB GPU (batch size = 16, one-step forward) using existing high-performance backends. Specifically, we utilize the SVDQuant Nunchaku (W4A4) [R1] kernels for 4-bit operations and native PyTorch (W8A8) for 8-bit operations. By applying our VaSMP and VaTMP strategies to allocate these hardware-supported bit-widths—adopting about **3:1 ratio** between **W4A4** and **W8A8**—we achieve a significant **~4.0x** end-to-end wall-clock speedup compared to the FP model, while the uniform W4A4 configuration yields a **~4.5x** speedup. This practical configuration demonstrates that our method can be effectively deployed using currently available high-performance kernels. As shown in the table below, it achieves superior image quality that surpasses the W4A4 setting and matches the high-frequency preservation of the W4A6 setting reported in our manuscript.
> > >
> > > | Metrics\Dataset | RealSR |
> > > | ----------------- | ------ |
> > > | LPIPS↓            | 0.3421 |
> > > | MUSIQ ↑           | 66.95  |
> > > | MANIQA ↑          | 0.4410 |
> > > | ClipIQA ↑         | 0.5272 |
> > > | LIQE ↑            | 3.466  |
> > >
> > > Regarding fully arbitrary-precision kernel support (e.g., W3A3/W5A5), we acknowledge that this remains an open engineering challenge.  While we view these specific configurations as a meaningful theoretical exploration of the intrinsic redundancy within DiT4SR, we are actively exploring the adaptation of our method to emerging arbitrary-/irregular-precision kernels (e.g., Quant-LLM  [R2] and ABQ-LLM [R3]) to bridge this gap. To support future research and facilitate better hardware adaptation, we will release our code and models soon.
> > >
> > > Overall, our work establishes a robust framework that bridges the gap between theoretical analysis and practical deployment, providing the community with a high-performance solution for accelerating DiT-based Real-ISR task without compromising perceptual fidelity greatly.
> > >
> > > Thank you again for the continued engagement. We are happy to address any further questions or concerns.
> > >
> > >
> > >
> > > **References:**
> > >
> > > [R1] Li, Muyang and Lin, Yujun and Zhang, Zhekai and Cai, Tianle and Li, Xiuyu and Guo, Junxian and Xie, Enze and Meng, Chenlin and Zhu, Jun-Yan and Han, Song. "SVDQuant: Absorbing Outliers by Low-Rank Components for 4-Bit Diffusion Models." ICLR 2025 Spotlight.
> > >
> > > [R2] Haojun Xia and Zhen Zheng and Xiaoxia Wu and Shiyang Chen and Zhewei Yao and Stephen Youn and Arash Bakhtiari and Michael Wyatt and Donglin Zhuang and Zhongzhu Zhou and Olatunji Ruwase and Yuxiong He and Shuaiwen Leon Song. "FP6-LLM: Efficiently Serving Large Language Models Through FP6-Centric Algorithm-System Co-Design." USENIX ATC24.
> > >
> > > [R3] Zeng, Chao and Liu, Songwei and Xie, Yusheng and Liu, Hong and Wang, Xiaojian and Wei, Miao and Yang, Shu and Chen, Fangmin and Mei, Xing. "ABQ-LLM: Arbitrary-Bit Quantized Inference Acceleration for Large Language Models."  AAAI 2025.

---

### Official Review · Reviewer_NhyH · 2026-03-05

**Soundness:** 3
**Presentation:** 3
**Significance:** 3
**Originality:** 4
**Overall Recommendation:** 4
**Confidence:** 3

**Summary:**

This paper presents a post-training quantization (PTQ) framework specifically for Diffusion Transformers (DiTs) used in real-world image super-resolution. While DiTs have shown state-of-the-art restoration quality, their deployment is often bottlenecked by immense computational costs. The authors identify that generic quantization methods for text-to-image DiTs-based generative models fail to preserve the sharp, high-frequency textures essential for SR.

To solve this, they introduce three main components:
   • H-SVD (Hierarchical SVD): Instead of a simple low-rank approximation, the authors decompose weights into global and local (block-wise) branches. This dual-structured approach is novel as it captures both broad structures and the fine-grained details that are usually lost during low-bit quantization.
   • VaSMP & VaTMP: A two-fold mixed-precision strategy. VaSMP handles weight bit-allocation across different layers without needing calibration data, while VaTMP dynamically adjusts activation precision over diffusion timesteps.

The most notable contribution is the achievement of stable SOTA performance even in aggressive W4A4 settings, where previous methods typically show significant quality degradation or generation collapse.

**Compliance With Llm Reviewing Policy:**

Affirmed.

**Final Justification:**

**Initial Assessment & Strengths/Weaknesses**: I found this paper to be technically sound. It introduces a good PTQ framework (H-SVD, VaSMP, VaTMP) for DiT-based Real-ISR. However, I had some concerns about (i) whether it works on other models rather than DiT4SR; (ii) if the claim on 60x speed-up can hold true in real applications, (iii) the technical reasons for visual errors (Figure 8), and (iv) whether the calibration is stable across different datasets.

**Rebuttal Evaluation**: The authors gave a clear and strong response that fully answered my concerns:

   • Generalization: Extra tests on DreamClear showed that the method works well on other models too.

   • Speedup Claims: The authors correctly changed the estimated speedup to 6.14x and shared actual end-to-end speed and memory results.

   • Artifacts & Limitations: The technical reason for the visual errors (weight/activation mismatch) makes sense. Their promise to add a dedicated "Limitations" section is a great improvement.

   • Calibration: New tests on DrealSR and DIV2K showed that the timing method is stable across different datasets.

**Final Recommendation**: The authors' reply cleared up my concerns. The paper makes an important contribution by making W4A4 quantization stable for DiT-based SR models. As long as the promised changes, corrected claims, and new tests are included in the final paper, I gladly maintain my **Weak Accept** rating.

**Key Questions For Authors:**

Questions about weaknesses:
   1. Generalization to other architectures: Your framework is primarily evaluated on DiT4SR. Given that different Real-ISR models (e.g., DreamClear) may have varying layer configurations, have you tested Q-DiT4SR on these models?
   2. Hardware-level efficiency: While you report a 60x reduction in theoretical operations, the branching nature of H-SVD and the dynamic switching of bit-widths in VaTMP could introduce hardware overhead. Can you provide empirical latency or FPS data on a target GPU (e.g., NVIDIA A6000)? Demonstrating actual speedup would validate the practical significance of this work.
   3. Analysis of texture preservation trade-offs: In Figure 8, adding H-SVD and VaSMP appears to cause texture collapse or color distortion in certain structural patterns (e.g., white lines) compared to the baseline. Could you explain the technical cause of these artifacts? Furthermore, how does the final Q-DiT4SR model (with VaTMP) mitigate these specific distortions?
   4. Robustness to domain shift: Since VaTMP relies on a small calibration set of 32 images, how sensitive is the temporal scheduling to the choice of these images? If the performance remains stable across different image domains, it will confirm the reliability of your "minimal calibration" claim.

**Limitations:**

No limitations mentioned in main paper:

While the authors briefly mention that there are no specific negative societal impacts to highlight, they have not adequately discussed the technical limitations of the proposed Q-DiT4SR framework in main paper. To improve the paper, the authors should include a dedicated "Limitations" section in main paper.
e.g.) Boundary Cases for H-SVD and VaSMP: As hinted by the subtle artifacts in the qualitative comparisons (e.g., in Figure 8), it remains unclear whether proposed components struggle with specific structural patterns or extreme textures. A deeper analysis of where the model underperforms would be highly valuable.

**Strengths And Weaknesses:**

Strengths:
   1. Effective Hierarchical Decomposition: The proposed H-SVD is a well-motivated approach that bridges the gap between global structure and local detail. By integrating a block-wise rank-1 branch, the model manages to preserve high-frequency textures that are typically destroyed in standard global SVD during aggressive quantization.
   2. Practical Mixed-Precision Strategies: The introduction of VaSMP and VaTMP offers a highly efficient way to handle bit-allocation. The fact that VaSMP is entirely data-free and VaTMP requires only minimal calibration makes this framework exceptionally deployment-friendly compared to existing PTQ methods.
   3. Significant Efficiency Gains: The empirical results are impressive, especially under the W4A4 setting. Achieving SOTA performance while reducing the model size by 5.8x and computation by over 60x demonstrates clear practical value for real-world, resource-constrained scenarios.

Weaknesses:
   1. Backbone Dependency: While the framework is presented as a solution for DiT-based Real-ISR, the evaluation is heavily centered on “DiT4SR”. To truly validate the "first systematic framework" claim, it would be necessary to see results on other architectures like “DreamClear” to ensure the H-SVD and variance-aware logic generalize well.
   2. Theoretical vs. Practical Speedup: There is a noticeable gap between the claimed 60x reduction in operations and actual hardware performance. Given the complexity of mixed-precision branching and H-SVD, theoretical GFLOPs may not translate to linear latency gains. FPS or latency benchmarks on the RTX A6000 would clarify the actual throughput benefits.
   3. Artifact Analysis in Intermediate Components: Figure 8 shows that H-SVD and VaSMP yields specific artifacts, such as color distortion in fine linear patterns (e.g., white lines), before VaTMP is applied. The paper would benefit from a deeper technical explanation of why these specific patterns are sensitive to these components.
   4. Calibration Robustness: Although the calibration set is small (32 images), Real-ISR is notoriously sensitive to domain shifts. A brief discussion or experiment on how VaTMP's temporal scheduling holds up when the test domain differs significantly from the calibration set would strengthen the paper's soundness.

---

> ### Author Rebuttal · Authors · 2026-03-30
>
> `Q2-1` Generalization: The method is mainly evaluated on DiT4SR. Does it generalize to other Real-ISR architectures (e.g., DreamClear)?
>
> `A2-1` We thank the reviewer for this suggestion. While our main experiments use DiT4SR, the method is not backbone-specific. To validate generalization, we further evaluate on **DreamClear**. Due to space constraints, we report results on one representative dataset (RealSR) here. Additional results on more datasets can be found in our response to **Reviewer jgvJ (`A3-3`)**. The results show consistent transferability, with W4A6 achieving competitive or even higher no-reference IQA scores than FP. Under W4A4, performance degrades as expected but remains competitive, supporting that our method is not restricted to DiT4SR.
>
> |  Metric   | DreamClear  (FP)| Q-DiT4SR  (W4A6) | Q-DiT4SR  (W4A4) |
> | :-------: | :--------: | :-------------: | :-------------: |
> |  LPIPS ↓  |   0.3249   |     0.3691      |     0.3752      |
> |  MUSIQ ↑  |   57.77    |      62.94      |      56.33      |
> | MANIQA ↑  |   0.4267   |     0.5060      |     0.4005      |
> | ClipIQA ↑ |   0.4695   |     0.5272      |     0.4308      |
> |  LIQE ↑   |   3.082    |      3.472      |      2.715      |
>
> We interpret the W4A6 improvements cautiously. A plausible explanation is that moderate quantization acts as a mild perceptual regularizer, suppressing unstable high-frequency details. This also highlights the limitation of existing no-reference metrics for quantized generative restoration.
>
>
>
> `Q2-2` Hardware efficiency: Given the branching in H-SVD and dynamic bit-width in VaTMP, can you provide empirical latency/FPS on a target GPU to validate practical speedup beyond theoretical reduction?
>
> `A2-2` We thank the reviewer for this important question. Due to space constraints, we have provided detailed analysis of hardware-level efficiency in our responses to **Reviewer DDAE  (`A1-1`)** and **Reviewer jgvJ  (`A3-1`)**. If any aspect remains unclear, we would be happy to further clarify and discuss.
>
>
>
>
>
> `Q2-3` Artifacts and limitations: What causes the texture/color artifacts observed in Fig. 8, how does VaTMP mitigate them, and what are the method’s limitations on challenging structures or textures?
>
> `A2-3`  We thank the reviewer for these helpful comments. We attribute the artifacts to the mismatch between **weight approximation** and **timestep-dependent activation quantization** in W4A4.  For layer $\ell$ and timestep $t$,
>
> $$
> \delta y_t^{(\ell)} = \left(W^{(\ell)} - Q\left(W^{(\ell)}\right)\right) a_t^{(\ell)} + Q\left(W^{(\ell)}\right)\left(a_t^{(\ell)} - Q\left(a_t^{(\ell)}\right)\right).
> $$
>
> H-SVD and VaSMP mainly reduce the former, while the latter accumulates across timesteps. Thin or high-contrast structures (e.g., white lines) are particularly sensitive, making small chromatic errors visually amplified.
>
> We validate this via a ROI study on random 32 RealSR SR images (one $64\times64$ region each) using ROI-PSNR, Edge MAE, and Color MAE, which are standard PSNR/MAE formulations applied to ROI regions, gradient space, and chromatic channels, respectively. Results show: H-SVD improves all metrics; adding VaSMP further improves Edge MAE but slightly degrades ROI-PSNR / Color MAE; and the full model with VaTMP achieves the best overall performance. This suggests that intermediate artifacts are mainly driven by **activation mismatch**, and VaTMP mitigates them by allocating higher precision to sensitive timesteps.
>
> | Method       | ROI-PSNR ↑  | Edge MAE ↓ | Color MAE ↓ |
> | ------------ | ----------- | ---------- | ----------- |
> | Baseline     | 19.6684     | 0.2757     | 3.7694      |
> | HSVD         | 19.8906     | 0.2739     | 3.5504      |
> | HSVD + VaSMP | 19.7209     | 0.2678     | 3.8786      |
> | **Q-DiT4SR** | **20.9590** | **0.2235** | **3.2286**  |
>
> More broadly, this reveals a technical limitation on thin, repetitive, or extreme textures. If accepted, we will add a dedicated **Limitations** section in the camera-ready version to explicitly discuss these cases.
>
>
>
> `Q2-4` Calibration robustness: How sensitive is VaTMP’s temporal scheduling to the choice of the small calibration set (32 images), and does it remain stable across different data domains?
>
> `A2-4` We thank the reviewer for this suggestion. We additionally tested the robustness of VaTMP by replacing the original 32-image RealSR calibration set with 32 images from DrealSR and DIV2K_train (random 128×128 crops), while still evaluating the resulting W4A4 model on RealSR. As shown in the table, the performance remains reasonably stable across calibration domains, with only minor variation rather than collapse.
>
> | Metrics\Calibration Set | RealSR | DrealSR | DIV2K  |
> | ----------------------- | ------ | ------- | ------ |
> | MUSIQ ↑                 | 66.36  | 66.24   | 65.92  |
> | MANIQA ↑                | 0.4367 | 0.4301  | 0.4228 |
> | ClipIQA ↑               | 0.4956 | 0.4963  | 0.4942 |
> | LIQE ↑                  | 3.179  | 3.296   | 3.035  |

---

> > ### Author Rebuttal · Reviewer_NhyH · 2026-04-03
> >
> > I thank the authors for their thorough and transparent rebuttal. The additional analyses and clarifications address my main concerns and strengthen my confidence in the paper. Assuming these corrections and clarifications are properly reflected in the final version, I maintain my Weak Accept recommendation and support acceptance.
> >
> > For the camera-ready version, I encourage the authors to incorporate the following points:
> >
> > 1. The additional DreamClear evaluation provides useful evidence that the proposed framework is not limited to DiT4SR. I also appreciate the authors’ cautious interpretation of the W4A6 improvements and their acknowledgment that W4A4 still degrades relative to full precision. Including these results in the supplementary material would strengthen the broader applicability claim.
> >
> > 2. The correction of the speedup claim should be reflected consistently throughout the manuscript. The rebuttal appropriately clarifies that the theoretical operation reduction is 6.14× rather than 60×, and the additional practical evidence, including the reported end-to-end speedup and memory reduction, is valuable enough to be included in the main paper or appendix.
> >
> > 3. The rebuttal’s artifact analysis is helpful and technically convincing. The explanation that the intermediate artifacts arise from a mismatch between weight approximation and timestep-dependent activation quantization clarifies why VaTMP is particularly important in the W4A4 regime. I strongly encourage the authors to add a dedicated Limitations section discussing failure cases on thin, repetitive, or high-contrast structures.
> >
> > 4. The additional calibration experiment is reassuring. Performance remains reasonably stable when the small calibration set is replaced with samples from DrealSR or DIV2K, which largely addresses the concern about domain shift. Including this comparison in the appendix would further support the claim that the method requires only minimal and reasonably robust calibration.
> >
> > Overall, the rebuttal directly addresses my main technical concerns with meaningful empirical evidence. I appreciate the authors’ thorough response and maintain my Weak Accept recommendation.

---

> > > ### Author Response · Authors · 2026-04-03
> > >
> > > We sincerely thank the reviewer for the positive evaluation and for acknowledging that our rebuttal has addressed the main concerns. We greatly appreciate the constructive feedback and the continued support for acceptance.
> > >
> > > We will incorporate all suggested improvements in the camera-ready version, including clarifying the speedup claims, adding the DreamClear results, expanding the artifact analysis with a dedicated limitations discussion, and including additional calibration experiments to further strengthen the paper.
> > >
> > > Thank you again for your helpful comments and support.

---

### Official Review · Reviewer_DDAE · 2026-03-12

**Soundness:** 3
**Presentation:** 3
**Significance:** 2
**Originality:** 2
**Overall Recommendation:** 4
**Confidence:** 4

**Summary:**

This paper proposes a novel Post-Training Quantization (PTQ) framework, named Q-DiT4SR, tailored for Diffusion Transformer (DiT)-based Real-World Image Super-Resolution (Real-ISR). The key issue addressed is the significant computational burden and large memory overhead of DiT models, which hinder their real-world deployment in image super-resolution tasks.

**Compliance With Llm Reviewing Policy:**

Affirmed.

**Final Justification:**

The author has addressed my concerns. I keep my initial concerns.

**Key Questions For Authors:**

1. Please compare the runtime and memory consumption in real-world scenarios with other methods.

2. This direction doesn't present many issues in the literature, but I feel that the increasingly complex quantization methods are disconnected from real GPU architectures and are hard to implement. In reality, the most useful methods are still the simple ones.

**Limitations:**

The authors adequately discuss the limitations, especially in terms of the challenges of aggressive low-bit quantization and the sensitivity of diffusion models to quantization errors.

**Strengths And Weaknesses:**

Strengths:

The novel combination of hierarchical weight decomposition (H-SVD) and mixed precision scheduling (VaSMP and VaTMP) for image super-resolution sets this work apart from previous quantization frameworks tailored to other models like U-Nets or text-to-image models.

Weaknesses:

The proposed methods, while effective, introduce complexity in terms of implementation and potential tuning. Although the authors minimize the calibration required, the approach may still require significant expertise to implement efficiently in various settings.

Please compare the runtime and memory consumption in real-world scenarios with other methods.

---

> ### Author Rebuttal · Authors · 2026-03-30
>
> `Q1-1` Please compare the runtime and memory consumption in real-world scenarios with other methods.
>
> `A1-1` We thank the reviewer for this important suggestion. We agree that practical runtime and memory should be evaluated in addition to theoretical compute reduction.
>
> We first clarify a numerical error in the manuscript. The Ops value of the FP model was mistakenly reported as **170,476**, while the correct value is **17,047.6** (rounded to **17,048**). The corresponding theoretical speedup should therefore be corrected to **17,048 / 2,776 ≈ 6.14×**. If accepted, we will correct these mistakes and provide the detailed calculation procedure in the camera-ready version.
>
> From an implementation perspective, our deployment is efficient (tested on RTX-4090 48GB, batch size = 16, one-step forward). The H-SVD global branch uses branch-parallel execution (similar to Nunchaku [R1]), while the local branch is implemented with an efficient Triton kernel [R2] inspired by Monarch matrices, resulting in minimal runtime overhead.
>
> Compared with **W4A4 SVDQuant** (~**4.8×** end-to-end speedup), our method achieves ~**4.5×** speedup with better visual quality, indicating that it is not merely a theoretical improvement. We also observe substantial memory reduction as shown in the table below.
>
> | Method   | Peak Memory (MiB) |
> | -------- | ----------------- |
> | FP       | 15085.99          |
> | SVDQuant | 3722.83           |
> | Ours     | 3974.64           |
>
> To better understand the runtime behavior, we further analyze the computation breakdown:
>
> | Component               | FP (ms) | Ours (ms) | Speedup |
> | ----------------------- | ------: | --------: | ------: |
> | Quantized Linear Layers | 1580.91 |    175.88 |   8.99× |
> | Nonlinear / Unquantized | 1518.39 |    510.62 |   2.97× |
>
> The gap between theoretical and end-to-end speedup is mainly due to the relatively limited acceleration of **nonlinear / unquantized components**, which become the dominant bottleneck. Importantly, the additional structure introduces minimal overhead: the **H-SVD branch** adds only ~**3%** latency, indicating that the efficiency is not hindered by the proposed design itself.
>
> Due to time constraints, we have not implemented native W3A3 / W5A5 kernels. However, our method does not rely on special operators and only assigns bit-widths to standard linear computations, making it  in principle compatible with existing arbitrary-/irregular-precision kernels (e.g., Quant-LLM [R3], ABQ-LLM [R4]). We will further explore adapting our method to such backends in future work.
>
> Overall, these results demonstrate that our method achieves a competitive balance of runtime, memory efficiency, and reconstruction quality in real-world deployment.
>
>
>
>
>
> `Q1-2`  This direction doesn't present many issues in the literature, but I feel that the increasingly complex quantization methods are disconnected from real GPU architectures and are hard to implement. In reality, the most useful methods are still the simple ones.
>
> `A1-2`  We agree with the reviewer that practical deployability on current GPU architectures is an important concern, and that simple quantization schemes are often the easiest to adopt in real systems. We do not view our work as contradicting this point. Instead, our goal is to explore whether more structured PTQ designs can improve the accuracy-efficiency trade-off for DiT-based Real-ISR, a setting where straightforward low-bit quantization often causes severe texture degradation. In this sense, we believe the contribution of our work is not only the specific design itself, but also the evidence that detail-preserving low-bit PTQ for DiT-based restoration can be pushed further than previously shown.
>
> More importantly, we see this line of work as creating possibilities for future practical deployment rather than claiming that every mixed-precision configuration is already directly supported by commodity GPUs. Our framework helps identify where precision matters most across layers and timesteps, which can inform future hardware-aware simplifications, kernel design, or more implementation-friendly approximations. We also agree that practical usability matters. For this reason, we aim to make the framework open-source and modular, so that future work can adapt it toward more hardware-friendly settings and simplified implementations.
>
>
> **References:**
>
> [R1] https://github.com/Nunchaku-AI/Nunchaku
>
> [R2] https://github.com/HazyResearch/fly
>
> [R3] https://github.com/usyd-fsalab/fp6_llm
>
> [R4] https://github.com/bytedance/ABQ-LLM

---

### Decision · Program_Chairs · 2026-04-30

**Decision:**

Accept (regular)

**Comment:**

The three reviewers all highlighted the strengths of this paper:

1. It explores post-training quantization for Diffusion Transformers (DiTs), which is very novel for Real-World Image Super-Resolution (Real-ISR).

2. The main components—Hierarchical SVD (H-SVD) and Variance-aware Spatio-Temporal Mixed Precision (VaSMP and VaTMP)—are well-motivated, and their effectiveness is validated through ablation studies.

3. The model reduction performance is promising.

The reviewers also raised several questions:

1. Can the proposed quantization method be applied to other DiT-based Real-ISR methods, such as DreamClear?

2. What is the wall-clock speedup?

3. How is performance affected if a different image set is used for VaTMP calibration?

4. Can the proposed method be executed on current-generation GPUs for arbitrary bit-widths (e.g., 3-bit, 5-bit)?

5. What are the limitations of the proposed method?

During the author-reviewer discussion period, the authors noted that they had incorrectly reported a key Ops value for the FP model. This error inflated the theoretical speedup ratio from 6.14x to 60x. After receiving this correction, I forwarded the information to all reviewers. The authors also clearly expressed the update in each rebuttal to ensure the reviewers are well-noted. One reviewer did not submit an acknowledgment, while another stated that their questions were fully resolved. The third reviewer indicated that their questions were only partially resolved but maintained an "Accept" recommendation regardless.

Ultimately, all three reviewers kept their scores unchanged, and all recommended the paper as "Accept" or "Weak Accept." Based on the novelty of the proposed method, the promising experimental results, and the clear explanations provided, the paper is recommended for acceptance.